# Student-Engagement Detection in Classroom Using Machine Learning Algorithm

Nuha Alruwais [1] and Mohammed Zakariah [2,*]

1 Department of Computer Science and Engineering, College of Applied Studies and Community Services, King Saud University, Riyadh 11451, Saudi Arabia
2 College of Computer Science and Information, King Saud University, Riyadh 11451, Saudi Arabia
* Correspondence: mzakariah@ksu.edu.sa

**Abstract:** Student engagement is a flexible, complicated concept that includes behavioural, emotional, and cognitive involvement. In order for the instructor to understand how the student interacts with the various activities in the classroom, it is essential to predict their participation. The current work aims to identify the best algorithm for predicting student engagement in the classroom. In this paper, we gathered data from VLE and prepared them using a variety of data preprocessing techniques, including the elimination of missing values, normalization, encoding, and identification of outliers. On our data, we ran a number of machine learning (ML) classification algorithms, and we assessed each one using cross-validation methods and many helpful indicators. The performance of the model is evaluated with metrics like accuracy, precision, recall, and AUC scores. The results show that the CATBoost model is having higher accuracy than the rest. This proposed model outperformed in all the aspects compared to previous research. The results part of this paper indicates that the CATBoost model had an accuracy of approximately 92.23%, a precision of 94.40%, a recall of 100%, and an AUC score of 0.9624. The XGBoost predictive model, the random forest model, and the multilayer perceptron model all demonstrated approximately the same performance overall. We compared the AISAR model with Our model achieved an accuracy of 94.64% compared with AISAR 91% model and it concludes that our results are better. The AISAR model had only around 50% recall compared to our models, which had around 92%. This shows that our models return more relevant results, i.e., if our models predict that a student has high engagement, they are correct 94.64% of the time.

**Keywords:** machine learning; deep learning; student engagement; student learning; virtual learning environment; classroom environment

## 1. Introduction

In the teaching–learning process, student engagement [1] is essential because it is closely related to learning rate. Here, the term, "student engagement" describes a measure of a student's level of interaction with others, plus the quantity of involvement in and quality of effort directed towards activities that lead to persistence and completion. It affects the learning process, encourages pupils to enhance their critical thinking abilities, and aids in retention. It is believed that encouraging student learning and growth [2] depends on their level of participation. Designing flexible, highly adaptive learning environments [3] that can accommodate a range of student engagement and learning preferences requires an understanding of how students interact with these technologies. Digital technology of all kinds is being rapidly incorporated into educational settings. By understanding how students use digital devices [4], educators can give students a variety of digital literacy skills and information to aid in their learning. Emotional, behavioural, cognitive, and agentic engagement are the four major areas into which student engagement is divided [5]. The behavioural component of student engagement refers to participation in extracurricular, social, and academic activities, as well as effort and perseverance. It mostly focuses on

participating in in-class events, finishing assigned work, and keeping a regular attendance schedule. Authors have [6] emphasized that good behaviour makes up the behavioural component of student engagement. The term "cognitive engagement" describes the psychological commitment to educational activities [7,8], where the learner is motivated to learn. Students who grasp a subject, show an understanding of it, and show a willingness to study and master skills exhibit this dimension. The cognitive form of engagement is associated with self-regulated learning [9], real questions about intellectual capacity, concentrating on activities, and setting objectives.

In an effort to reduce the number of students who are expelled, methods for measuring and analysing student engagement in learning have been actively developed. In modern classrooms, MOOCs [10], educational games, intelligent teaching systems, etc., managing student engagement levels is important. One of the crucial elements that should be emphasized in education is student participation [11]. The term "student participation" describes a student's performance in a course outside of their assessments. Here, the items that might be evaluated include engagement in class discussion, engagement in online discussion, and behaviour in group settings. The level of student engagement is frequently linked to academic success and can be used to address disciplinary issues in schools. Collaboration between school administrators, instructors, and parents [12] is essential in creating relevant activities that not only place an emphasis on academic accomplishment but also foster students' socio-emotional growth [13] and help them comprehend the proper meaning of involvement in the classroom. The teaching–learning dynamic can be better understood in light of student engagement. In order to respond to shifts in their students' motivation, involvement, and attitude towards their courses and academic goals [14], instructors might alter their instructional practices by measuring the level of student engagement.

In e-learning systems, it is important to be able to predict the students' level of participation because it enables the teacher to see how each student interacts with the course's many activities. In adaptive learning environments, student engagement prediction is also helpful for improving the functionality of recommender systems. The current work aims to determine the most effective machine learning (ML) method [15–18] for predicting student engagement as well as understanding the most effective features for predicting in virtual learning environment (VLE) courses. Classification techniques are often used in learning analytics. The prediction model seeks to determine whether a student is highly or minimally interested in the platform and how VLE activities affect student engagement in the platform [14].

We established a predictive analytical model to assist us in accomplishing our objectives using ML approaches. Due to the correlation between low student engagement and high failure rates, the best ML prediction model was chosen to analyse student participation in VLE classroom instruction and gauge students' levels of participation in VLE courses. By determining which assignments and materials are more crucial to the course evaluation, teachers utilize statistical methods to assist students in succeeding in the course [18,19]. These models also give educators the chance to engage students in a variety of VLE activities, encouraging them to take the VLE course. Instructors must identify the reasons why a student is losing patience in a certain activity or section of the course material. Tracking student engagement in different educational learning activities improves overall learning, and a comprehensive review of student participation can reduce the number of course dropouts. Our models are simple to implement into VLE systems and can allow teachers to identify students who are not engaging, through the use of various exams and various course materials. This is the first study to use AI to forecast student involvement in a VLE course. To examine how students engage with VLE courses, we used a range of ML algorithms as analytical learning approaches and compared the results [20].

This study uses extracted raw data to compute and describe student engagement. The most common data pre-processing techniques for student engagement detecting models are converting the format of input data by normalization, encoding. The prediction process is followed in this work by applying models like XGBoost [20], LightGBM [21], random

forest [22], and CatBoost [23], and then, finally, the stacking technique. Here, we create a deep learning model with two crucial steps—basic data-input detection and engagement recognition—to overcome the issues. To create a rich input representation model and achieve cutting-edge performance, a convolutional neural network (CNN) is trained on the dataset in the first step. This model is used to initialize our engagement-recognition model, which was created using a different CNN and was trained using our recently amassed dataset in the engagement recognition domain.

Almost all web-based learning management systems for classrooms offer straightforward access to behavioural traits. The term "behavioural traits" in the context of student engagement refers to the observable acts of students who are being involved in learning. It also refers to students' participation in academic activities and efforts to perform on different tasks. These characteristics also point to student engagement in a way that is more like a real-world task in a conventional learning environment.

The conceptual framework for student participation is shown in Figure 1. Data gathering and analysis come first. Data preprocessing, feature selection, classifiers, and finally data transformation and classification. The pupils are then divided into high- and low-engagement students by the evaluating and testing procedure, which employs a variety of techniques and a prediction model.

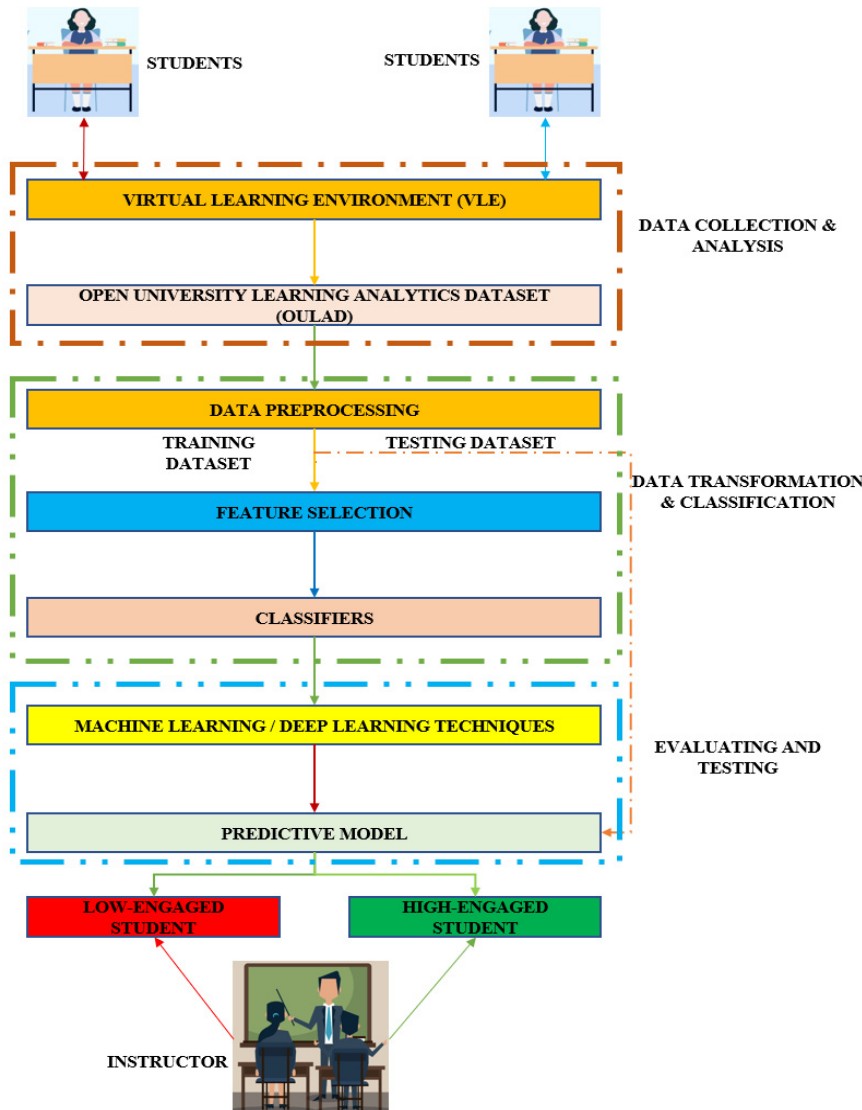

**Figure 1.** Conceptual Structure for Student Engagement Using ML/DL Methods in a Classroom.

In the conventional educational system [23], teachers encourage students to participate in their learning activities using a variety of methods, such as by having them pay attention, take notes, ask questions, and actively participate in class activities. Nowadays, with the introduction of technology-enhanced learning systems in the classroom [24], engagement tactics for learning are evolving. Therefore, increasing student engagement in online-learning environments [25] is still an open research subject that needs to be investigated.

This study is an early investigation into how data-driven interventions can improve student engagement in an online-learning environment. The level of student participation in this study is assessed using statistical data. Retention and attention are major important criteria in this study in which the investigation reveals the level of attention among the students. Activity logs are unbiased statistics that show how pupils actually behave when they are learning. We applied supervised random forest and unsupervised clustering machine-learning algorithms to analyse the correlations. The methods discovered an intriguing pattern in student engagement and demonstrate that both engagement and assessment results are reliable indicators of students' academic success. Preliminary research on student action and interactions, then enhance, and customize their techniques and content is among the other contributions of this study. Researchers have been able to address specific issues relating to students' learning processes and their effectiveness with the aid of machine learning algorithms. Additionally, the expectation-maximization (EM) [19] clustering algorithm is often employed in machine learning to cluster data. For a variety of reasons, this method has been used with educational data.

In order to forecast student engagement and participation in a VLE, this study compares various machine learning algorithms. Teachers can intervene in the learning process, even at an early stage of the courses, by using the accurately projected results. It is challenging for the models to provide accurate predictions due to the abundance of model parameters and the reliability of their properties. Therefore, the current study places more emphasis on the variables that help with the early prediction of the VLE engagement. The project also seeks to identify the best classifier for handling the varied, heterogeneous data from the VLE log. The following research queries are addressed in this work:

- Which classifier performs best in predicting students' VLE engagement?
- What are the variables that affect a VLE's ability to forecast student engagement optimally?

In order to predict a student's performance and engagement based on interactions, participation in class, and a variety of other variables, this study investigates the application of classification methods and other deep learning algorithms for predictive analytics. The paper's remaining sections are organized as follows: Section 2 discusses the literature review and Sections 3 and 4 describe the dataset and the methodology, respectively. Sections 5 and 6, respectively, offer the findings and conclusion.

## 2. Literature Review

There has been a great deal of research on student engagement in traditional educational settings and classroom-learning systems. A range of methodologies and input components have been used to study the relationship between student data and student engagement. The authors of [14] conducted a study to learn more about student engagement in blended-learning courses in higher education. Using a cross-lagged modelling technique, the study found that certain factors affecting student engagement and course design had a substantial impact on student involvement in the course. Using a statistical approach, Ref. [26] investigated the link between a student's interest in the subject and their final grade. Students who actively engaged in the reading and quizzes functioned better in the final exam, researchers noted. Employing engagement-related input elements to establish an early-warning system, Ref. [27] observed that these parameters are strong predictors of issues with student engagement. Using a ML approach based on students' facial expressions and head movements, Ref. [28] evaluated student engagement. Their

findings revealed that ML algorithms were efficient at projecting student involvement in the classroom.

Researchers have recently used statistical methods to investigate the impact of teacher presence, user-friendly circumstances, and academic self-efficacy on student engagement in MOOCs. Student involvement increased as the exam got closer, according to [29], who used data from the LMS platform to evaluate student engagement and performance statistics. Furthermore, they identified a connection between academic achievement and student interest. In [30], investigation has shown that student responses to online-learning activities have a significant impact on test outcomes. SVM and KNN classifiers are effective measures of student participation, according to the analysis by [31] of student involvement using ML.

Similar to this, Ref. [32] employed a variety of ML methods to analyse the Student Performance Dataset and the Student Academic Performance Dataset, using the former dataset for prediction and the latter for categorization. Students' online behaviours were taken into consideration by [33] to forecast their success while utilizing an e-learning system. Using information gleaned from students' log-in histories and use of a learning management system on the Sakai platform, the author classified students according to their learning styles [33]. Preprocessing, feature selection, and parameter optimization were completed before classification. The performance of students in a particular course can be predicted with the use of this kind of categorization. Another study [34] has shown that ML methods may accurately predict a student's final grades by using their prior grades. A dashboard was created to predict students' engagement and performance in real-time, which could help stop students from deciding to drop out too early. In a different study [35], ML methods were employed to predict students' engagement on the basis of their behavioural characteristics and to examine how evaluation marks were affected. With the use of a dashboard that shows students' behaviours in the learning environment, instructors can quickly detect pupils who are not paying attention in class. An adaptive gamified learning system [32–34] was created to boost student engagement in the learning environment and, as a result, their performance. Such a system integrates educational data mining with gamification and adaption approaches. In the context of online learning, the effectiveness of gamification in comparison to adaptive gamification was examined. Three classifiers were used by [34] to create a framework for predicting student engagement and performance. By eliminating unnecessary and duplicate elements, the authors preprocessed the data obtained from the Kalboard 360 online-learning management system. They then carried out feature selection and analysis to determine which features were the most discriminating. Finally, classification techniques were employed to forecast the performance of the students.

In higher education, classroom instruction is becoming more common and has been rising steadily. Numerous academics [34–36] have looked into the effects of technology on student engagement and learning outcomes as a result of the increasing popularity and usage of online learning. The majority of research in the field has claimed that information technology has a beneficial effect on student engagement. Some research suggests that asynchronous learning helps students develop higher-order abilities, such as analysis, synthesis, problem solving, judgement, simulation, and teamwork. Students who participate in online learning tend to collaborate more, according to [36]. The authors of [36,37] have highlighted that, because online learning has made higher education more accessible, there is now a need for more accountability and proof of learning efficacy. Student involvement is one of the most important markers of successful online learning. To ensure that this kind of teaching is effective, it is crucial to focus on student participation in an online classroom. There are numerous frameworks for classroom education that take technological advancements into account. A method for adaptive e-learning that makes use of content tracking and eye tracking has been presented by [38]. These researchers have provided a conceptual framework for spotting confusion in interactive digital learning settings to help learners clear up their doubts.

The three degrees of decisions had accuracy levels of 91.74% for not engaged, 89.55% for generally engaged, and 95.69% for highly engaged. Another study [39] looked at student engagement by analysing the trial results of five distinct CNN models. In [40] analysed the primary data-mining technique used in numerous projects. This classification article examined a method for modelling and forecasting student performance and engagement measures. On the other hand, very few studies have combined classification and clustering. Three of the most popular data mining methods include decision trees, Naive Bayes classifiers, and ANNs. Studies that analyse machine learning models on the VLE log frequently concentrate on data acquired from students enrolled in the course at a certain point in time. Data exploration is a difficult task because of the large search space [39].

Figure 2 shows the methods used in various studies on how to keep students engaged in the classroom. In these studies, student extraction takes the form of task data, actions, rating comments, and statistics. Also included as machine learning or deep learning models are XGboost, LightGBM, random forest, neural networks, CATBoost, and stacking methods. The model evaluation also includes the MOOC, VLE, CNN, EM, and AISAR models. The prediction results can be shared with the lecturer and the students at the end of the process.

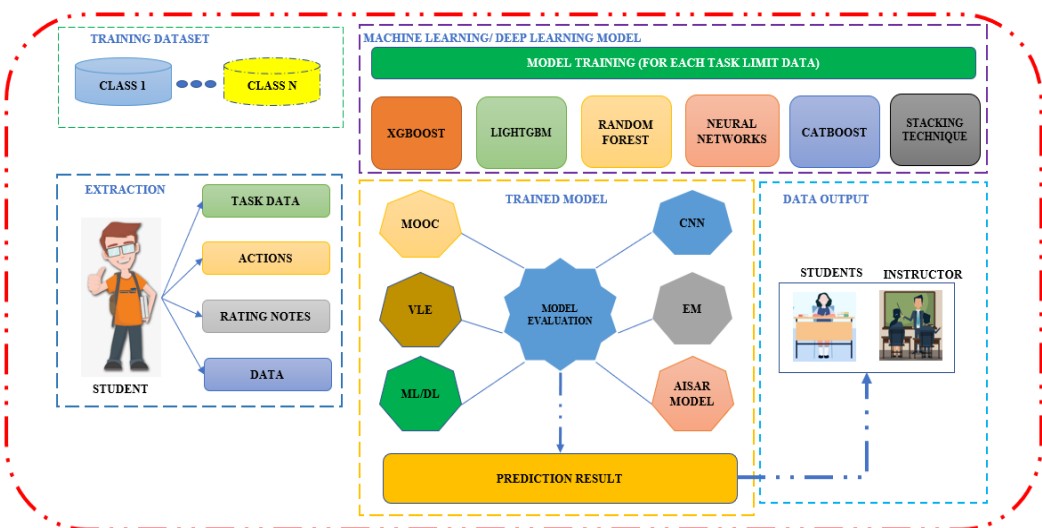

**Figure 2.** Techniques applied in different papers related to student engagement.

The Open University Learning Analytics Dataset (OULAD)[41], which has been anonymised, was used for this investigation. Further, the K-means+ clustering technique are applied [42]. In Ref. [43] applied statistical methods to predict student engagement in a web-based learning environment and concluded that variables such as course design, teacher participation, class size, student gender, and student age need to be controlled for when assessing student engagement. Ref. [43] conducted a study to understand student engagement in higher education blended-learning classrooms. This study used a cross-lagged modelling technique and found that course design and student perception variables greatly affected student engagement in the course. Ref. [44] conducted a study to investigate the relationship between a student's final score and the student's engagement in material using a statistical technique and found that students who had high levels of engagement in quizzes and materials earned higher grades in the final exam. Ref. [45] developed an early-warning system using engagement-related input features and found that these variables are highly predictive of student-retention problems. In [46] measured student engagement using a ML algorithm based on students' facial expressions, head poses, and eye gazes. Their results showed that ML algorithms performed well at predicting student engagement in class.

A list of earlier articles on the subject is included below in Table 1. These studies describe the approach employed, the features, the level of engagement, the results or outcomes, the model's accuracy, and the environment.

**Table 1.** List of Past Paper References with Methodology Used and Results.

| Ref | Dataset | Methodology | Features | Engagement | Outcome | Accuracy | Environment |
|---|---|---|---|---|---|---|---|
| [1] | 63 surveys, 9 interviews, 16 participants | SPSS software, deep learning, learning analytics data | Qualitative interview | Student engagement, involvement in learning, academic student experience | Behavioural engagement, engagement in online services | 75% of survey participation | Classroom |
| [5] | 28,710 images of faces, validation sets include 3570 images | CNN, Haar-Cascade Classifier, AdaBoost, MES Dataset, FER 2013 | Student performance evaluation | Student engagement, engagement probability | Improving digital learning method | Model accuracy, 93.6%, 98.9%, and 88% | e-learning |
| [14] | 32,593 students participated | LSTM model, learning analytics | Logistic regression and artificial neural networks | Online educational system, student engagement | Open University learning analytics dataset | Accuracy of 97.25% and precision of 92.6% | Online classroom learning |
| [10] | Total of 11 students interviewed | Virtual learning environment, LMS tool, adaptive hybrid model | Interview data | Student engagement and perception | Massive open online courses | Accuracy of 91.45% | e-learning |
| [23] | 1105 registered first-year full-time students were analysed | Moodle, Microsoft Excel, dataset, data collection | Histogram, distribution and performance values | Student engagement level | Activity-based learning, course engagement | Engagement level of 0.991 | Classroom environment and e-learning |
| [15] | 694 undergraduate students were analysed | KNN, SVM, ANN, decision tree, logistic regression, data mining, ML technique | Learning analytics | Student performance and engagement | Highest precision for student engagement data | Accuracy of 94.56% | Classroom |
| [16] | 788 students were recorded, 650 from Portuguese class, 395 from maths class | Educational data mining, ANN, VLE; deep learning technique, data-driven decision-making technique | Classification accuracy | Student academic performance | Facilitating decision making process towards future education | Accuracy of 85–90%, logistic regression accuracy of 80–84% | Classroom and e-learning |
| [32] | 137 students were analysed | Facial action units, SVM, regression model, deep learning, coding procedure | Classroom management, machine-vision-based approach | Student engagement, cognitive engagement | Pilot classroom recordings, knowledge test | Accuracy of 93.67% | Classroom |

**Table 1.** *Cont.*

| Ref | Dataset | Methodology | Features | Engagement | Outcome | Accuracy | Environment |
|---|---|---|---|---|---|---|---|
| [35] | 4093 students took part in the experiment of which 2580 failed the test | Machine learning and data mining techniques, C++ learning analytics, classification and regression algorithms, LMS model | Decision tree, single-feature and mixed-feature profiles | Student engagement and performance | Best performing classification models | Accuracy of 95.31% | Classroom and online environment |
| [36] | Selection process started from 292 research articles. 11 scientific articles. | ML algorithms, decision tree, naïve Bayes, SVM, random forest, KNN, logistic regression, dataset and analysis | Maximum likelihood estimation, random subset | Students' performance and engagement | Predicting student performance | Performance accuracy: 90%; NB accuracy: 81%; RF Accuracy: 28% | Classroom |

Numerous works have been produced concerning this area of student participation. The most common learning environment is a physical classroom or an online classroom. To extract variables such as student performance evaluation, student engagement, cognitive engagement, and student performance from the datasets, various techniques were used, and each of these approaches produced results with a varying level of accuracy. In these publications, writers compared their models to earlier models created by other authors, and in most cases, the accuracy of their models outperformed that of their competitors. Similarly, in this present work, our models demonstrate strong prediction performance with high accuracy, precision, recall, and AUC scores overall, with a slight advantage being shown by the CATBoost model. Our created model beat earlier studies in every way. In this study, we contrast several models, namely XGBoost, RF, MLP, and, lastly, the AISAR model, which was created with greater overall accuracy.

## 3. Dataset

The purpose of this study is to create the best predictive model for categorising students according to their participation in a VLE. The input is student data, and the algorithm should be able to forecast whether a student would have a high or low engagement level.

The Open University Learning Analytics Dataset (OULAD) [41], which has been anonymized, was used for this investigation. The Open University, a British public university, is home to the vast majority of undergraduate students in the UK. It is the largest educational institution in the UK, having enrolled two million students since its foundation in 1969 (and one of the biggest in all of Europe). The majority of Open University students attend classes off-campus, as suggested by the name. The dataset details seven selected modules (courses) and how students interacted with the virtual learning environment (VLE).

The course lectures, resources, and assessment data are all stored in the VLE. The students use the VLE to read texts, watch lectures, and finish tasks. Finally, log files that document and save student interactions with the VLE are maintained. The logs' descriptions of student behaviour include details such as their interactions with the VLE.

The dataset contains the following seven parameters: studentInfo, studentAssessment, assessment, studentVLE, studentRegistration, VLE, and courses. CSV files with these parameters are accessible. Figure 3 displays the parameters for the OULAD framework. Data on the students' demographics, registration, assessments, and VLE interactions are included in the dataset. In the assessment table, the learner performance is broken down into four categories: distinction, pass, fail, and withdrawn. Each student's results and accomplishments during the course are reflected in their performance as a learner. Utilizing interaction data from the VLE, the current study concentrates on the level of student engagement.

The dataset includes evaluations from module presentations that are followed by final exams, as well as information on seven chosen courses, which are referred to as modules in the dataset. The dataset also contains information about the students' location, age, disability, education level, gender, and other factors. Additionally, results from student evaluations and their interactions with the online-learning environment are given (VLE). The first step is collecting and analysing the data. Data transformation and classification come after data preprocessing, feature selection, then classifiers. Then, using a number of tools and a prediction model, the evaluating and testing procedure separates the students into high- and low-engagement groups. Moreover, many strategies for maintaining student engagement in the classroom have been used in this research. The information on the tasks, behaviours, rating remarks, and statistics are taken from the students' engagement. Additionally, XGBoost, LightGBM, random forest, neural networks, CATBoost, and stacking techniques are included as deep learning models. The AISAR model is also included in the model evaluation. In the end, the hoped-for result was achieved.

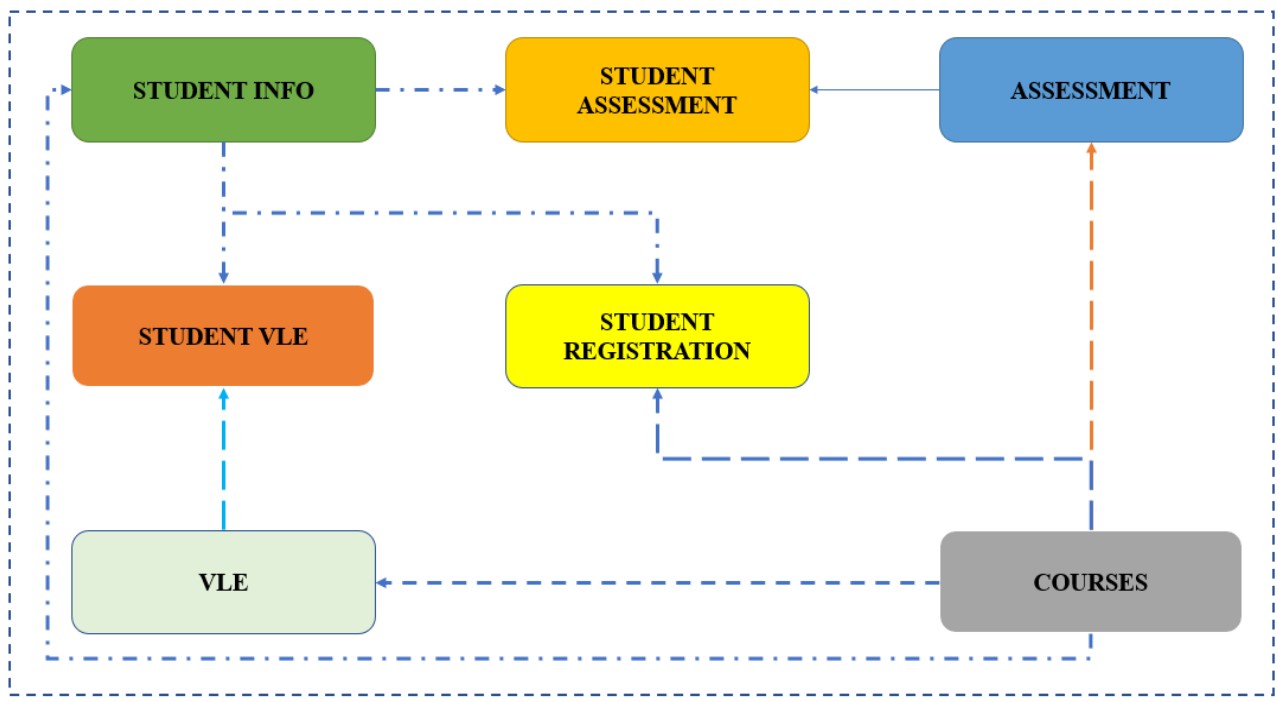

**Figure 3.** OULAD framework.

## 4. Methodology

The workflow followed for the development of the predictive model of student engagement is summarized in Figure 4. Data preparation and exploration was the initial step, which entailed preparing and examining the dataset to make it acceptable for model creation and to better understand its traits and patterns.

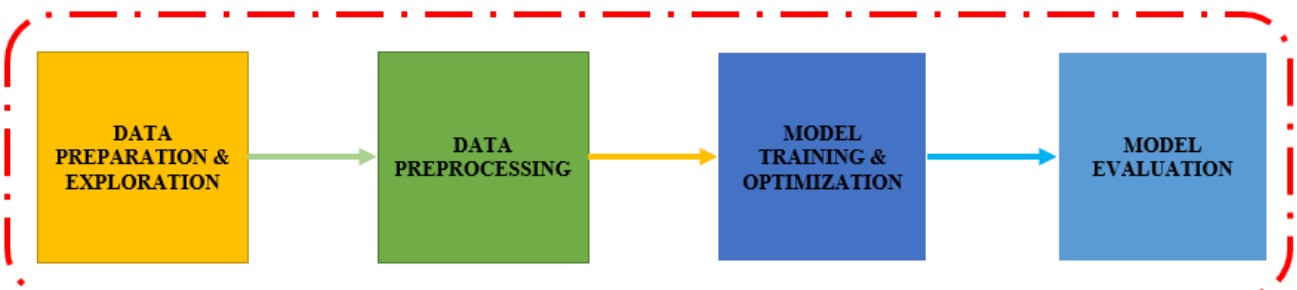

**Figure 4.** Model development workflow.

The following phase was data preprocessing, which involved cleaning and preparing the data for analysis. The third step, which involved training and testing many models using various techniques and hyperparameter settings, came after a thorough study and preprocessing of the dataset. Finally, the performance of each model was compared and evaluated using various metrics and graphs.

### 4.1. Data Preparation

Raw databases are often not in a form that is suitable for analysis, so it is common to perform some initial preparation on the data before proceeding with further analysis. The dataset used for the predictive model was cleaned of unnecessary attributes, which would not impact the prediction of the output, such as the identification of the student, the site, the assessments, and the presentations, as well as other irrelevant attributes such as the length of the course and the type and duration of the assessments.

The final dataset contained 13 features and 32,593 observations, with 8 features being categorical and 5 being numerical. The features used are listed below in Table 2, and their distribution is cited in the annex:

**Table 2.** The features selected for the training of the model.

| Name of the Feature | Description |
| --- | --- |
| Code_module | The identification code for the module. |
| Gender | The gender of the student. |
| Region | The geographic region of the student. |
| Highest_education | The highest level of education the student had at the time they entered the module. |
| Imd_band | The Index of Multiple Depravation bands [2] of the student's residence at the time the module was presented. |
| Age_band | The age interval of the student. |
| Num_of_prev_attempts | Number of previous attempts by the student for this module. |
| Studied_credits | The number of credits for the modules that the student is enrolled in. |
| Disability | Whether the student has declared as disabled or not. |
| Sum_click | The quantity of interactions the student has with the material. |
| Date_registration | The date on which the student registered for the module presentation. |
| Module_presentation_length | The number of days for the presentation of the module. |
| Starting_month | The month in which the student started the module. |

The target for prediction was an engineered variable that shows the engagement of the student that was used in [19]. This variable's value was 1 if the student has high engagement, and 0 if the student has low engagement. The dataset is slightly imbalanced, with approximately 72% of the students having low engagement, and 28% of the students being highly active. Distinction, qualification, and active participation during the course are the foundations of student engagement. The findings show that high-engagement students are more likely to complete evaluations with high marks (excellent) or to pass final exams while participating actively in VLE courses (active). Figure 5 depicts the distribution of the target variable of student engagement.

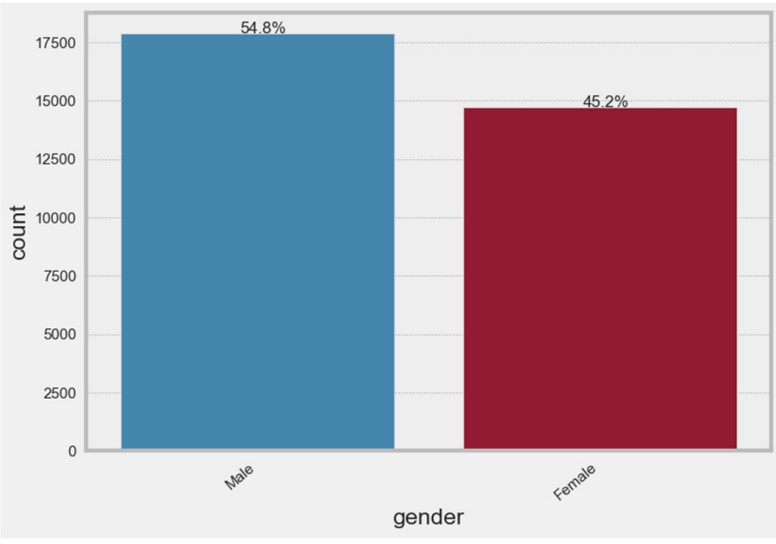

**Figure 5.** The Distribution of The Target Variable of Student Engagement.

### 4.2. Data Preprocessing

Data preprocessing is the process of preparing a dataset for analysis by cleaning, converting, and formatting it. Due to its potential to dramatically affect the accuracy and dependability of the analysis' outputs, this is a crucial phase in the data science process. Moreover, data loss can occur for a variety of reasons, including incorrect data entry, device failures, lost files, and more. Many statistical and ML approaches are not built to handle missing values. If missing values are not appropriately handled, they might produce biased or erroneous findings.

### 4.3. Missing Values

Missing values happen when specific variables do not have any data recorded. Data loss can occur for a variety of reasons, including incorrect data entry, device failures, lost files, and more. Every dataset typically contains some missing data.

For a variety of reasons, missing values can be problematic in machine learning. One issue is that many statistical and machine learning approaches are not built to handle missing values, and if missing values are not appropriately handled, they might produce biased or erroneous findings. Missing values can also render a dataset incomplete, which makes it more challenging to identify the connections between variables and derive actionable conclusions from the data.

The final dataset contained only a few missing values, since most of them were imputed during the preparation of the final dataset. The remaining missing values were imputed using either the mean for the numerical features or the most frequent attributes for the categorical features.

### 4.4. Normalization and Encoding

Normalization is a data preprocessing technique that scales the values of numerical features in a dataset to a common scale without affecting variations in the values or the connections between the features. Normalization is a crucial step in the preprocessing of data because it can assist machine learning algorithms perform better by making the data more receptive to them. Since many machine learning algorithms are sensitive to the scale of the input characteristics and can perform badly if the features are not on a common scale, this is frequently helpful when dealing with these algorithms.

The process of transforming categorical (i.e., non-numeric) data into a numeric format that machine learning algorithms can employ is known as categorical data encoding. Since many machine learning algorithms are built to operate with numeric data and cannot directly handle categorical data, this is important.

The normalization of the dataset was conducted following the type of attributes:

For the categorical features, one-hot encoding was performed to ensure the categories were viewed as independent of each other by the ML model. This technique for encoding categorical data generates a new binary feature (0 or 1) for each distinct category. The feature gender with two categories (male and female), for instance, would result in the creation of two new binary features, one for each category, gender_male and gender_female.

For the ordinal features, the features were encoded by mapping each unique label to an integer value. For example, the age band category was converted by order:

(0–35): 1, (35–55): 2, and (55<=): 3

For the numerical data, the features following probabilistic distribution like a normal distribution were normalized using the Z-score method, such that the mean of all the values was 0 and the standard deviation was 1.

For the numerical data that have varying scales, the associations in the original dataset were preserved by using the min-max normalizing approach, which scales the values of each feature to a predetermined range (for example, 0 to 1), and unit norm scaling, which scales the values of each feature so that the product of its squared values is 1.

The flowchart for the full list of actions needed for this method is shown in Figure 6. To begin with, we created the best prediction model for categorising student activity in a VLE.

Following data preprocessing, OULAD was measured using the learning environment. It was separated into normalization approaches and missing values. The classifier and feature selection are the results of data preprocessing. A predictive model was then produced during the evaluation and testing phase.

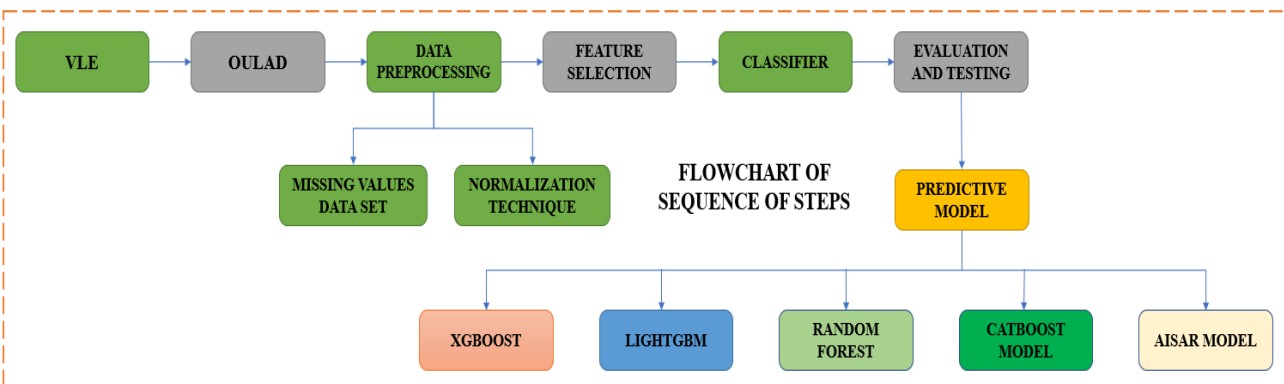

**Figure 6.** Flowchart of Sequence of Steps.

*4.5. Model Training and Optimization*

Several models were trained and their hyperparameters were optimized to be able to learn how to predict the engagement of the students. Different algorithms were used for the development of the predictive models: XGBoost [20], LightGBM [21], random forest [22], CatBoost, stacking techniques, and, finally, neural networks with multilayer perceptions, as well as other relatively weak models.

XGBoost (eXtreme Gradient Boosting) is a machine learning technique for creating ensemble models by repeatedly adding weak models to a strong base model. To create a stronger model, XGBoost trains several small decision-tree models individually. XGBoost has several features that make it powerful, such as fast training times, high scalability and flexibility, and high accuracy.

LightGBM is another ensemble machine learning model like XGBoost. The sole distinction is that trees grow leaf-wise in LightGBM, whereas they grow depth-wise in XGBoost, as shown in Figure 7.

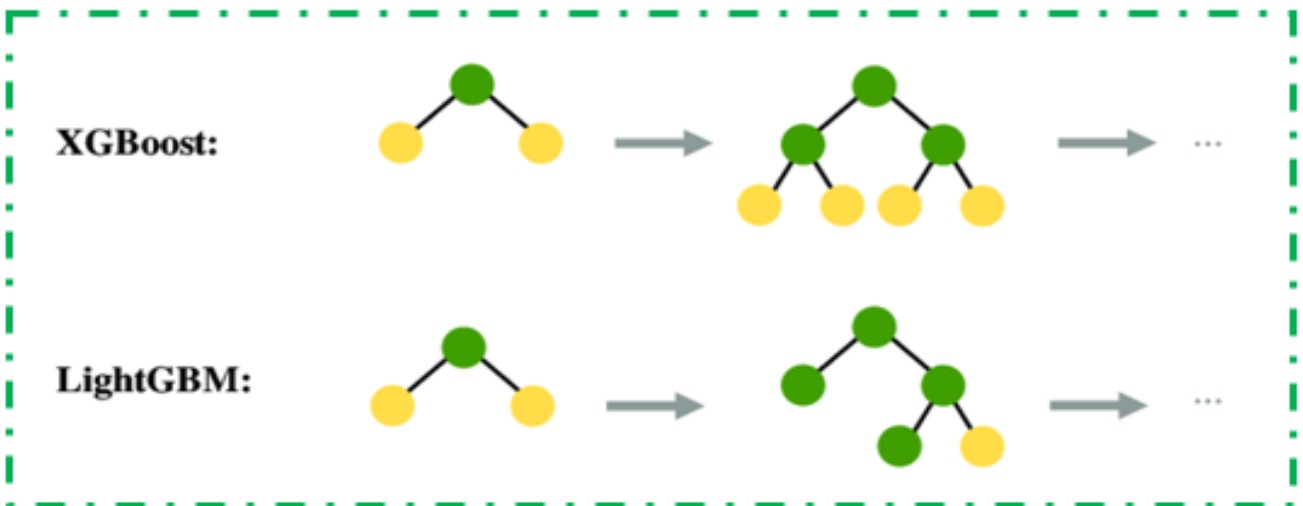

**Figure 7.** Difference between XGBoost and LightGBM.

Random Forest is a group of decision trees that are assembled to form an ensemble machine learning model. It is a type of bootstrapped ensemble, which means that it was

created by combining the predictions of several decision trees that were trained on various subsets of the data. Figure 8 shows the RF classifier architecture.

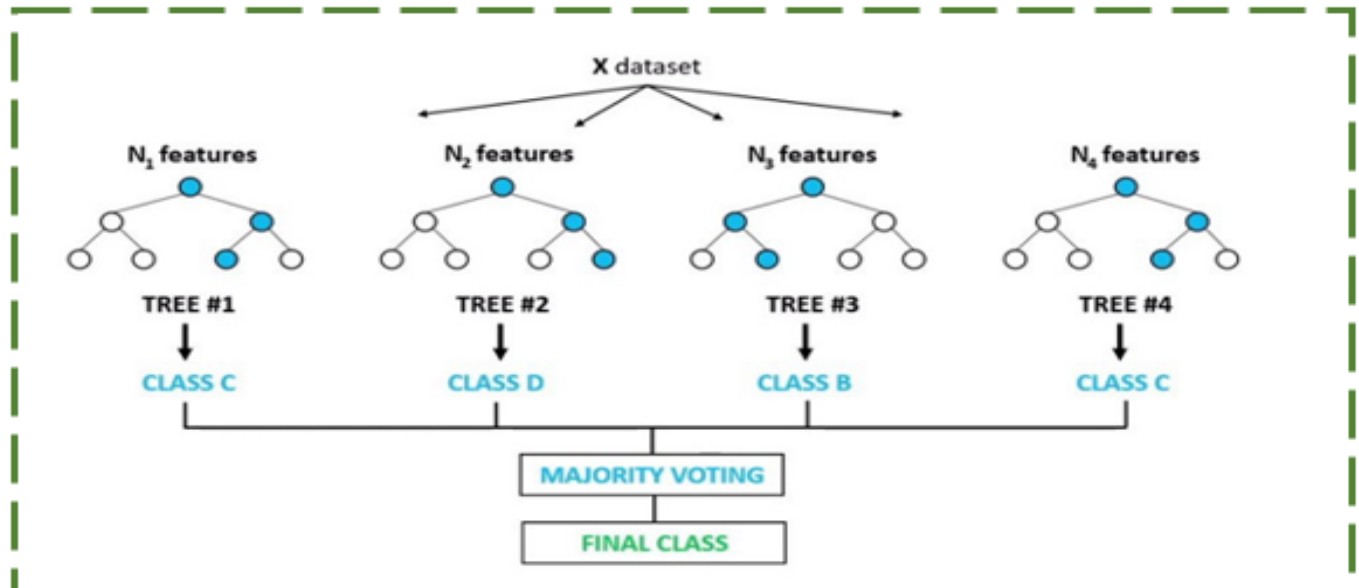

**Figure 8.** Random Forest Classifier Architecture.

CatBoost is a technique for decision trees that uses gradient boosting. The model excels at handling categorical variables. CatBoost can handle categorical data directly, without the requirement for preprocessing or encoding, in contrast to many other gradient-boosting tools. This makes it an easy tool to use when working with datasets that include a lot of categorical variables, like our dataset.

In contrast to XGBoost and LightGBM, CatBoost constructs symmetric (balanced) trees. The same condition is used to separate the leaves in each phase from the preceding tree. For each node in the level, the feature-split pair that causes the least loss is chosen and applied. This balanced tree architecture facilitates effective CPU implementation, reduces prediction time, produces quick model implementers, and inhibits overfitting due to the structure's regularization. Figure 9 depicts the CATBoost symmetric tree.

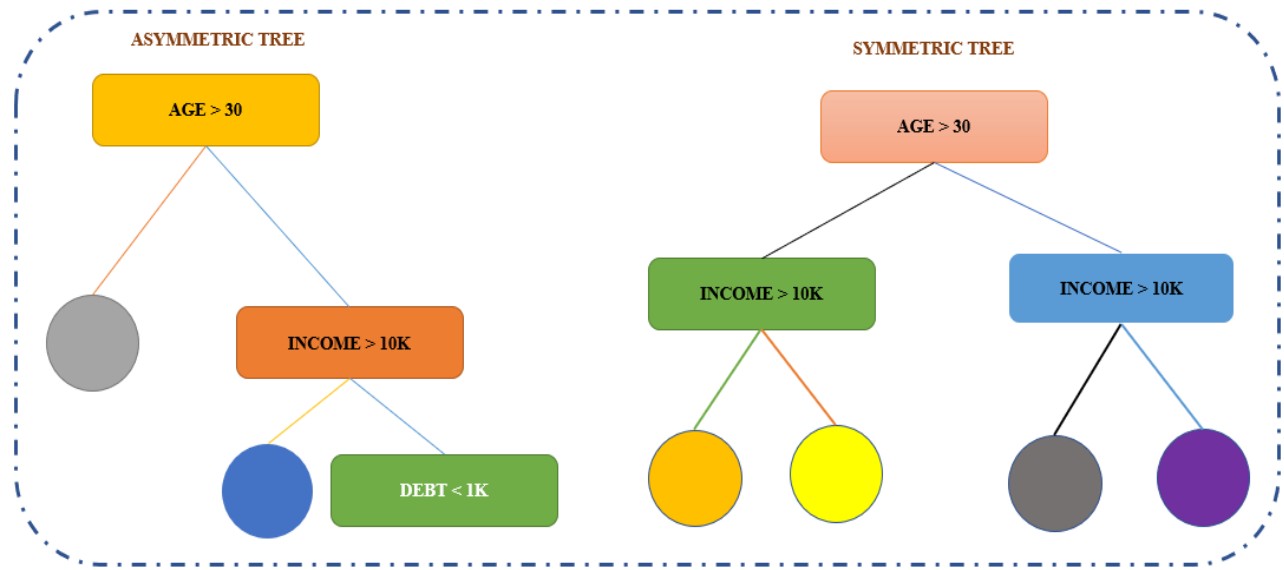

**Figure 9.** Catboost Symmetric Tree.

To enhance the performances, stacking techniques were applied to all of these models after they had each been trained independently. Stacking is a machine learning ensemble strategy that involves training a number of base models, utilizing their output predictions to inform the final prediction made by a higher-level meta-model. In order to create a stronger and more accurate overall model, stacking aims to integrate the advantages of the basic models.

After preprocessing the dataset, the one-hot encoding resulted in an increase in the number of features to 34. The large number of these features could have a negative impact on the predictive model's performance, because some of them might not be important for predicting the students' engagement and should be eliminated.

A subset of a dataset's features was chosen through the feature selection process to be used in analysis. This can help to simplify the analysis and boost the effectiveness of machine learning algorithms. During the training of each model, permutation-based feature importance was applied, which is a technique for assessing the significance of specific features in a dataset by determining how much the performance of a ML model is affected when the values of the feature are randomly permuted. This method can be used to determine which features are most crucial for forecasting the target variable. It can also be helpful for deciding which features to include in a model or for figuring out how a model makes predictions.

Feature engineering was also performed to manipulate and transform the raw data features into features that are well-suited for the predictive model. Using mathematical arithmetic operators such as addition, multiplication, and other operations, the golden feature search technique was utilized to create new features. Applying the decision-tree technique, which is a tree-like simple machine learning model used for classification that is used to make predictions by following the decisions made at each node in the tree, only the most crucial features are kept after evaluating the prediction value of newly formed features.

By minimising the objective function, the K-means+ clustering technique [42] leverages the concept of K-means clustering based on cosine distance to cluster the features so that the resultant feature subset has strong correlation and no redundancy:

$$J = \sum_{j=1}^{k} \sum_{i=1}^{n} \left|\left| x_i^{(j)} - c_j \right|\right|^2$$

$k$ is the number of clusters, $n$ is the number of observations, $c_j$ is the cluster centroid, and $i$ is the observation, where $J$ is the objective function.

To create a well-tuned and optimized model for every machine learning task, it is crucial to optimize the hyperparameters of the trained models. The best hyperparameters for each model were found using the Optuna software framework, which automatically optimizes hyperparameters. The log loss metric, also known as cross-entropy loss, is a metric used to assess a machine learning model's performance, particularly for classification tasks. All of the models were tuned using this metric. It calculates the gap between the true labels of the data and the expected probabilities of the model.

The multilayer perceptron design of the neural network used for training consisted of four layers: one input layer, two hidden layers, and one output layer. The input layer was 16 units thick (16 input features with 18 features dropped). The first hidden layer was composed of 32 units followed by a ReLU activation layer, in charge of activating the node or output for the input by converting the node's summed weighted input. A dropout layer was then applied to prevent overfitting by randomly dropping nodes during training. The second hidden layer was similar to the first hidden layer, with the same number of units, activation function, and dropout rate. The output layer consisted of a single unit and a sigmoid activation function, used for binary classification methods. Figure 10 shows the architecture of the multilayer perceptron.

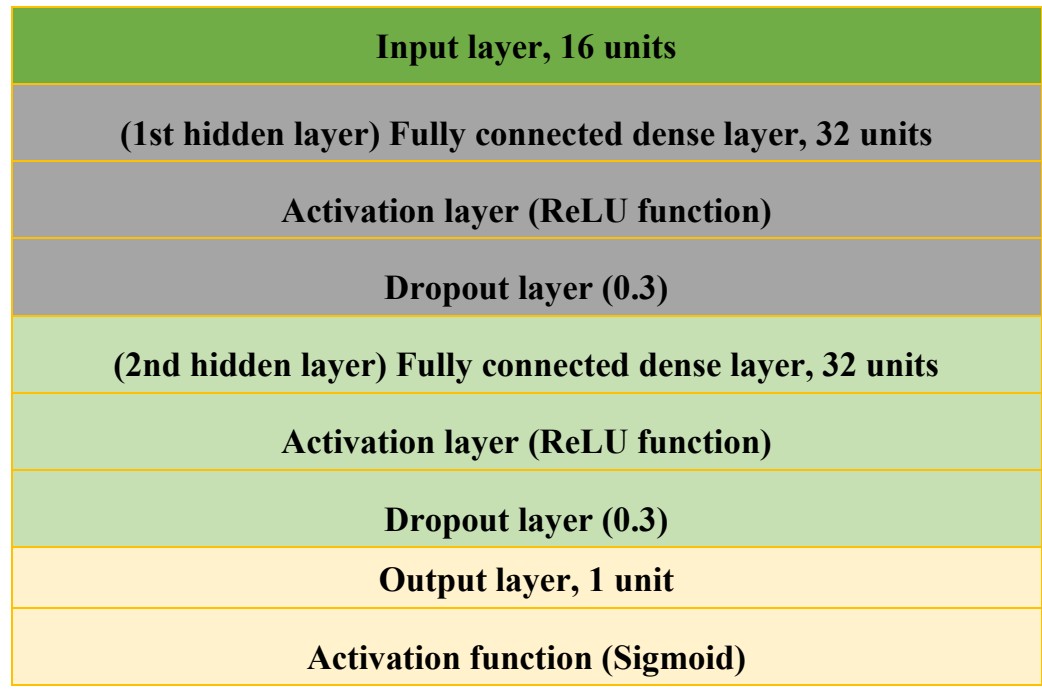

**Figure 10.** Architecture of the Multilayer Perceptron.

On the other hand, Figure 11 shows the fully connected representation of the neural network architecture, where a fully connected layer refers to a neural network in which each neuron applies a linear transformation to the input vector through a weight's matrix. As a result, all possible connections layer-to-layer are present, meaning that every input of the input vector influences every output of the output vector.

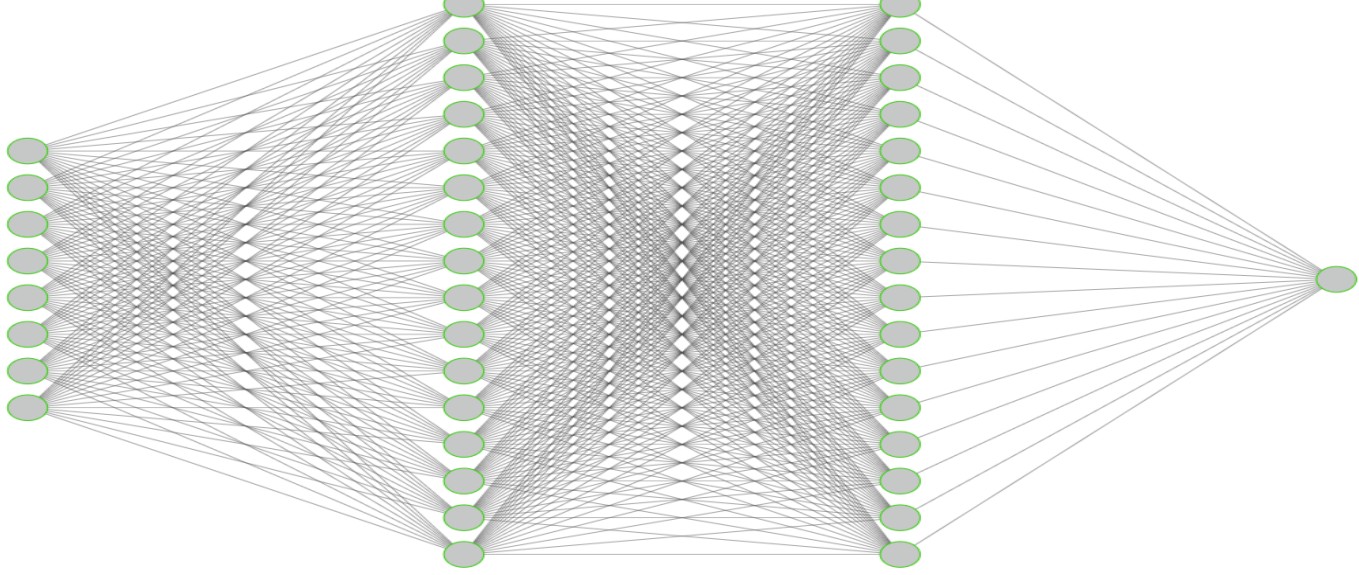

**Figure 11.** Fully Connected Representation of The Neural Network Architecture (Number of Units Divided By 2 For Visibility).

## 5. Results

The goal of the project was to create a model that can forecast a student's level of engagement in a VLE, from high to low. A portion of the OULAD dataset was used for the investigation, and the characteristics were chosen using correlation measurement. The dataset was created by pulling out and combining various columns from seven tables

that were made accessible through OULAD. The resulting dataset had a row for each student. According to the assessment score considered, as described in the section above on data preprocessing, the rows were then marked as high-engagement or low-engagement students. The algorithms discussed earlier in this paper have already shown a correlation between evaluation scores during the course and final scores with engagement.

In order to measure how well a machine learning model can predict future data, model assessment evaluates the model's performance on a collection of data. Additionally, compared to past research [19,42], this would allow us to evaluate how well it predicts students' participation in e-learning.

In addition to the log loss metric, other metrics, such as the accuracy, recall, precision, and f1 scores, were utilized to assess the model. The recall is calculated by dividing the total number of positive cases in the dataset by the proportion of true positive predictions. A high recall score indicates that the model can accurately identify the majority of the positive examples in the dataset, or all the students who exhibit high engagement. The precision is calculated by dividing the total number of positive predictions made by the model by the number of real positive predictions. A high precision score means that the model is very unlikely to provide many false positive predictions. The AUC is a crucial measurement in model evaluation that is also used to compare the effectiveness of various models. The receiver operating characteristic (ROC) curve, which compares the TPR against the FPR at various classification thresholds, is used to determine the AUC score. An excellent ability to discriminate between positive and negative classes is indicated by a high AUC score. Table 3 displays the metrics comparison between the trained models.

**Table 3.** Metrics comparison between the trained models.

| Model | Log Loss | AUC | Accuracy | Precision | Recall |
|---|---|---|---|---|---|
| LightGBM | 0.21606 | 0.9624 | 0.9223 | 0.9440 | 93.21 |
| RandomForest | 0.2206 | 0.9606 | 0.9218 | 0.939 | 92.78 |
| CatBoost | 0.2154 | 0.9626 | 0.9223 | 0.9464 | 92.3 |
| XGBoost | 0.2170 | 0.9623 | 0.9210 | 0.9458 | 93.2 |
| MLP | 0.2424 | 0.9507 | 0.9175 | 0.9362 | 91.9 |

*5.1. Algorithm for Optimal Classification*

Which classifier performs the best in order to predict student participation in a VLE? This is the very first question that was posed in the introduction section of this paper, and it is the focus of this results section. We can see that a variety of data sources were used to create the predictive models. The assessment result, the end result, and the students' selection of VLE activities are the features in this case. The degree of student engagement is the anticipated variable.

Likewise, the second question stated in the introduction section focused on student engagement, with the homepage, outgoing content, subpage, URL, and forum as a subset of activities that exhibits more encouraging outcomes. The result reported here is consistent with past research using datasets like OULAD and others. When compared to a student who was less engaged, a highly engaged student was found to interact with these activities more. Additionally, the authors were able to find a correlation between the results of these activities' clicks and their scores. In order to anticipate at-risk pupils, it is possible to use the interaction with these activities. Recall is the main metric used to determine which students are least involved, and accuracy is used to assess how well the model predicts which students will be most engaged. The figures that follow show the normalized confusion matrix and the ROC curve.

Figure 12 shows the normalized confusion matrix for the LightGBM model, while Figure 13 depicts the ROC curve for the LightGBM model. The best values for the training parameters for the LightGBM model were as follows: lambda l1 = 1.5, lambda l2 = 1,

learning rate = 0.01; min data in leaf = 50; num boost round = 2000; and reg alpha = 0.1. The LightGBM prediction model had an AUC score of 0.9624, a precision of 94.40%, a recall of 100%, and an accuracy of around 92.23%.

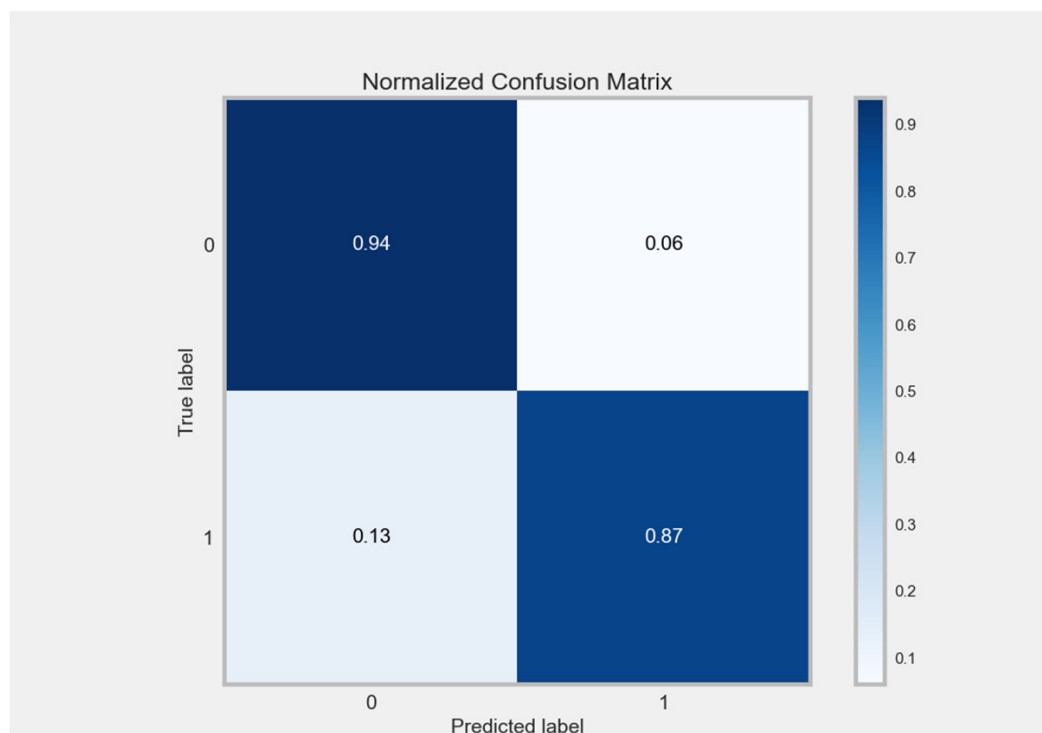

**Figure 12.** Normalized Confusion matrix for LightGBM model.

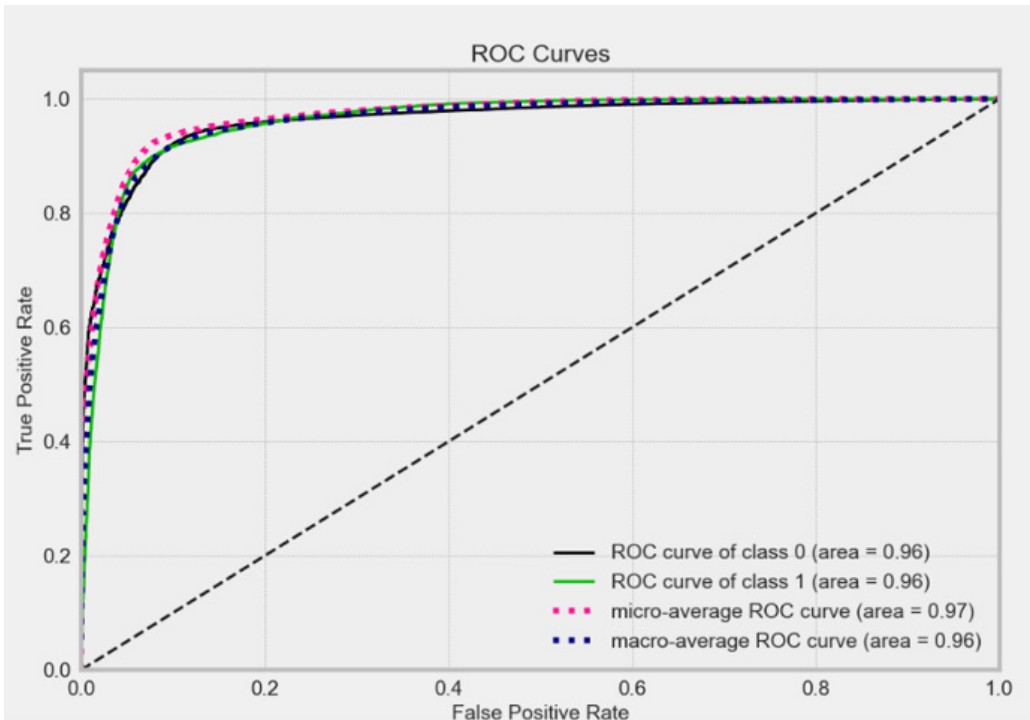

**Figure 13.** ROC Curves for LightGBM model.

Figure 14 shows the precision–recall curve for the LightGBM model, while Figure 15 depicts the normalized confusion matrix for the XGBoost model. The optimum parameters

we used to train the LightGBM and XGBoost models were max depth = 3, learning rate = 0, and n estimators = 140.

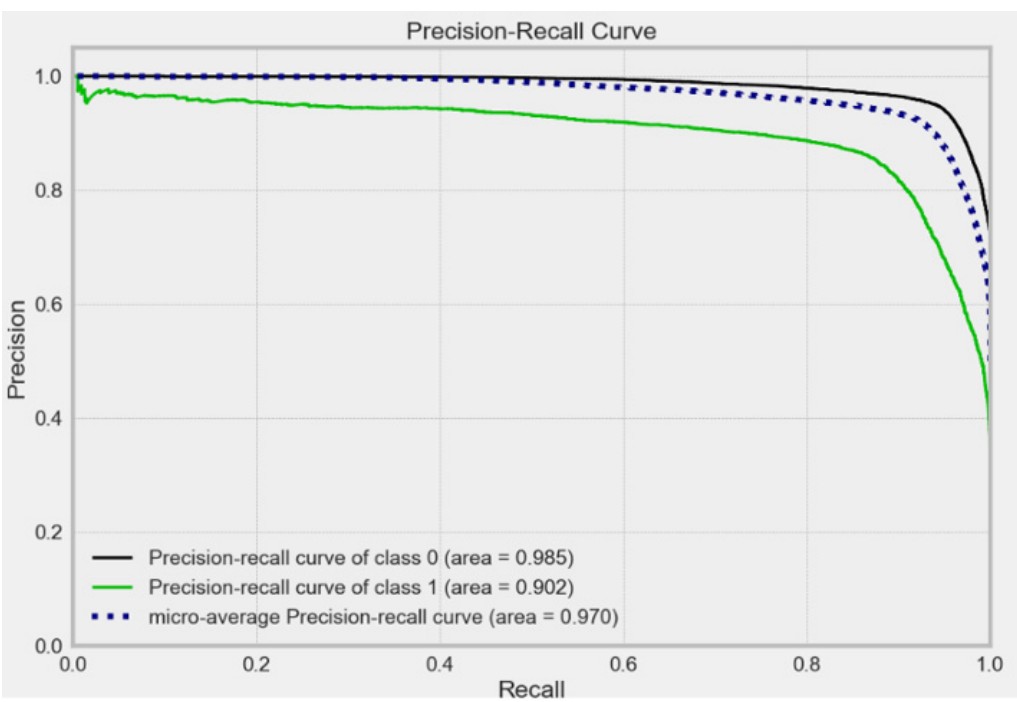

**Figure 14.** Precision–recall Curve for LightGBM model.

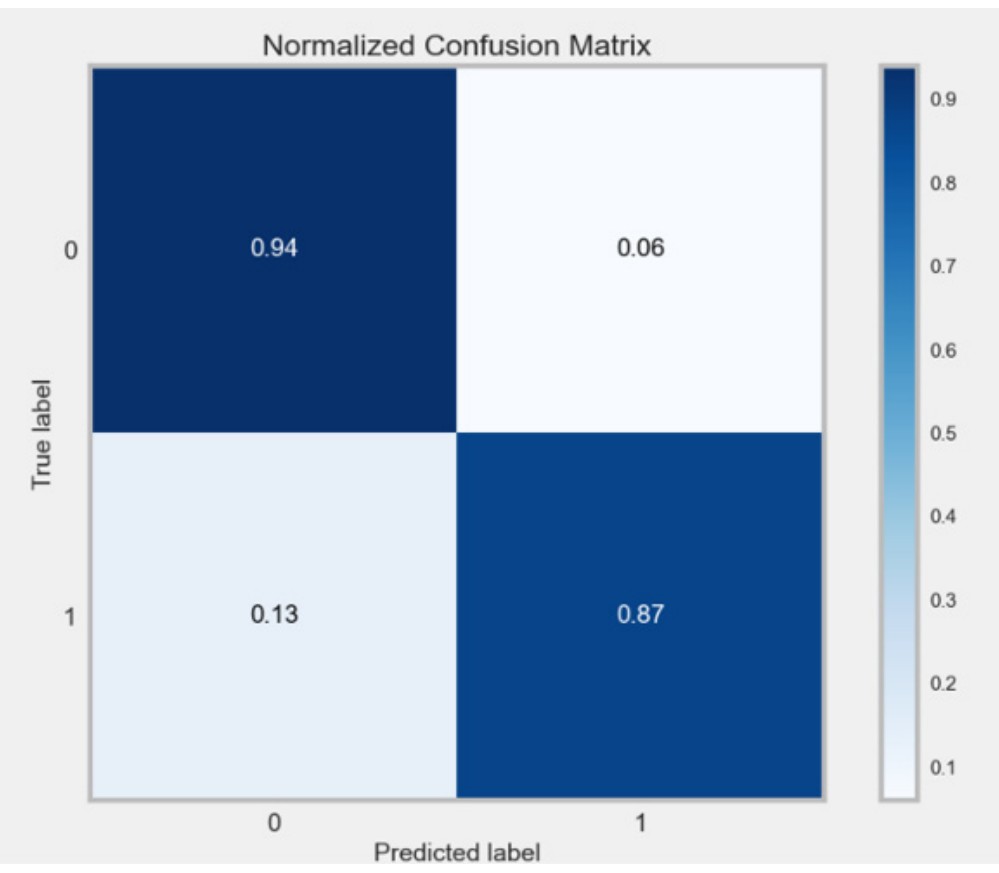

**Figure 15.** Normalized Confusion matrix for XGBoost model.

Figure 16 shows the ROC curve for the XGBoost model, while Figure 17 depicts the precision–recall curve for the XGBoost model. The XGBoost prediction model had an AUC score of 0.9623, a precision of 94.58%, a recall of 100%, and an accuracy of around 92.10%.

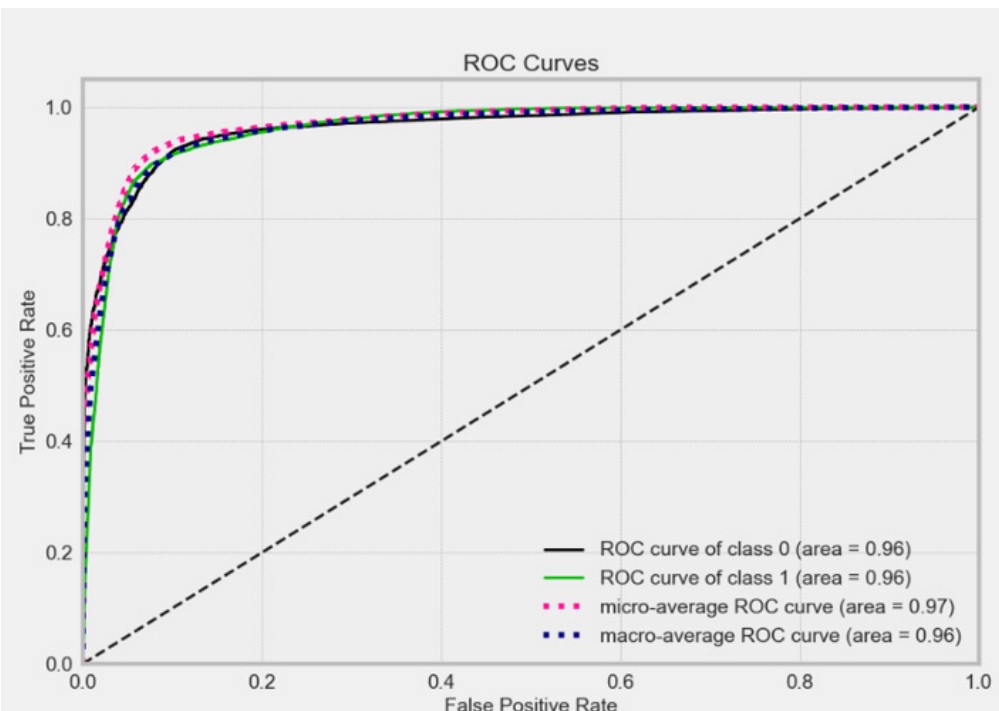

**Figure 16.** ROC Curves for XGBoost model.

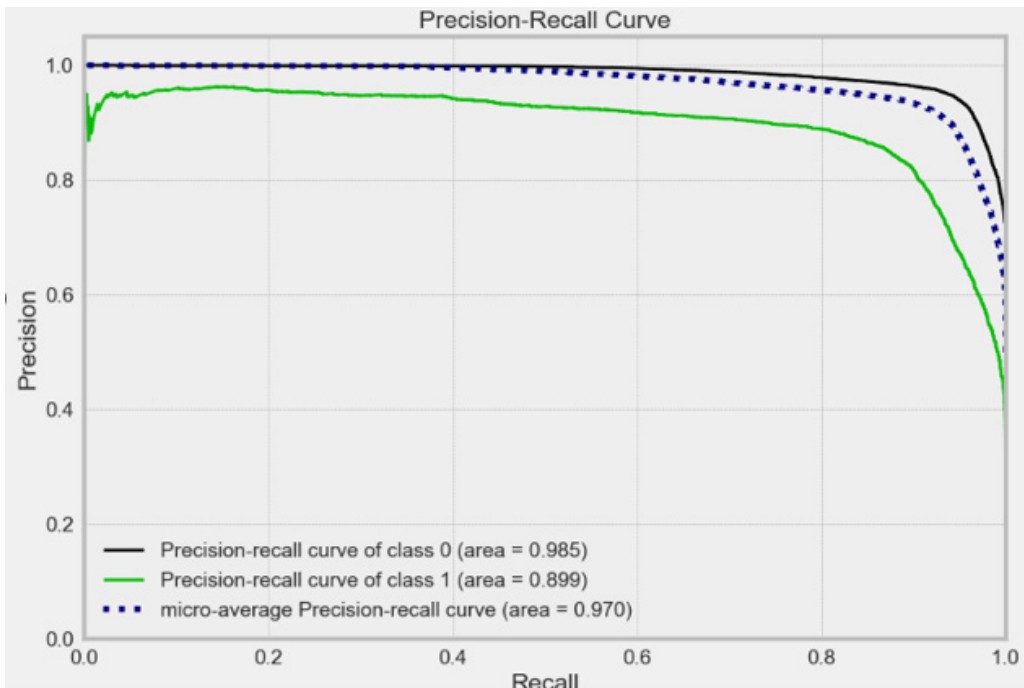

**Figure 17.** Precision–recall Curve for XGBoost model.

Figures 18–20 demonstrate the random forest model for normalized confusion matrix, ROC curves and precision–recall curve. The RF model has a log loss of 0.2206, AUC score of 0.96, accuracy of around 92.18%, precision of 93.9%, and recall of 92.79%.

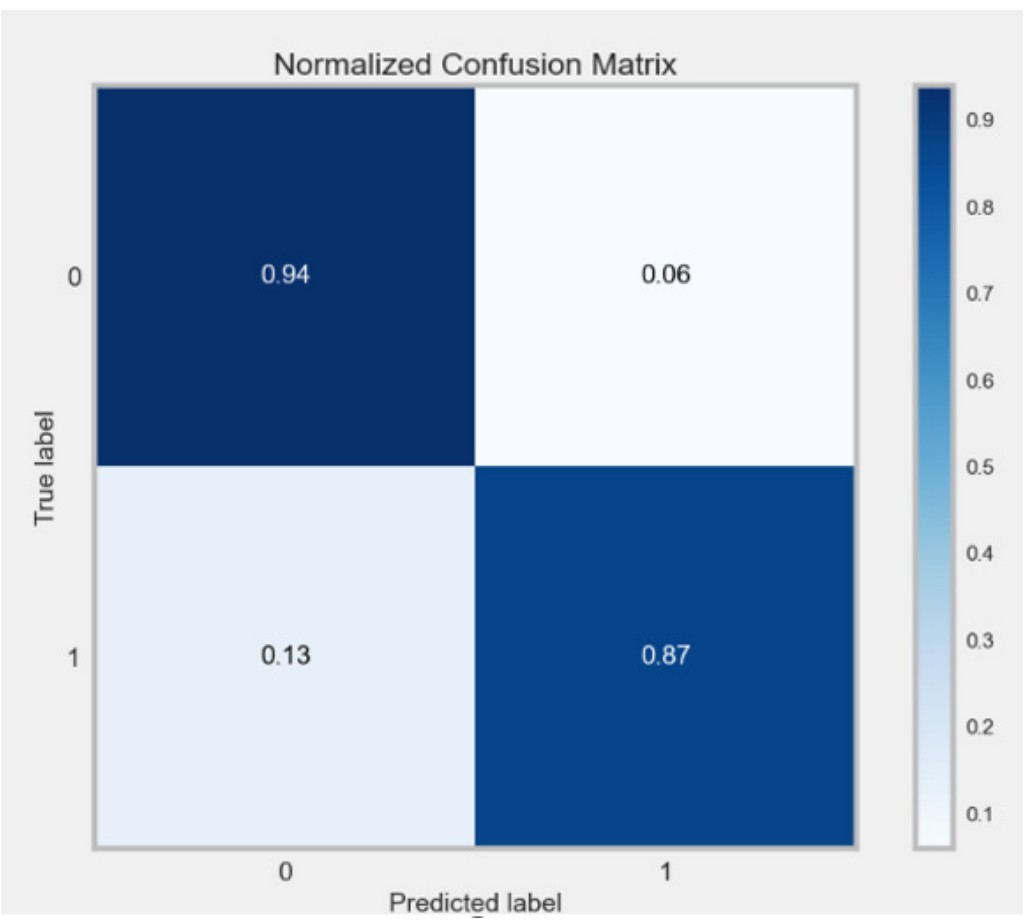

**Figure 18.** Normalized Confusion matrix for random forest model.

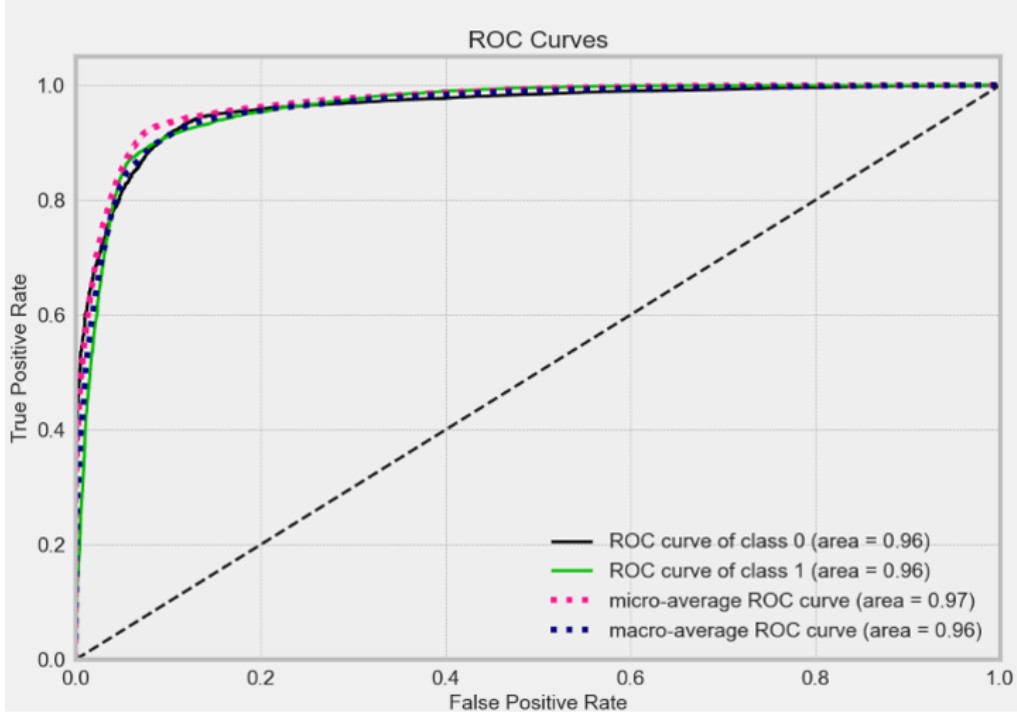

**Figure 19.** ROC Curves for random forest model.

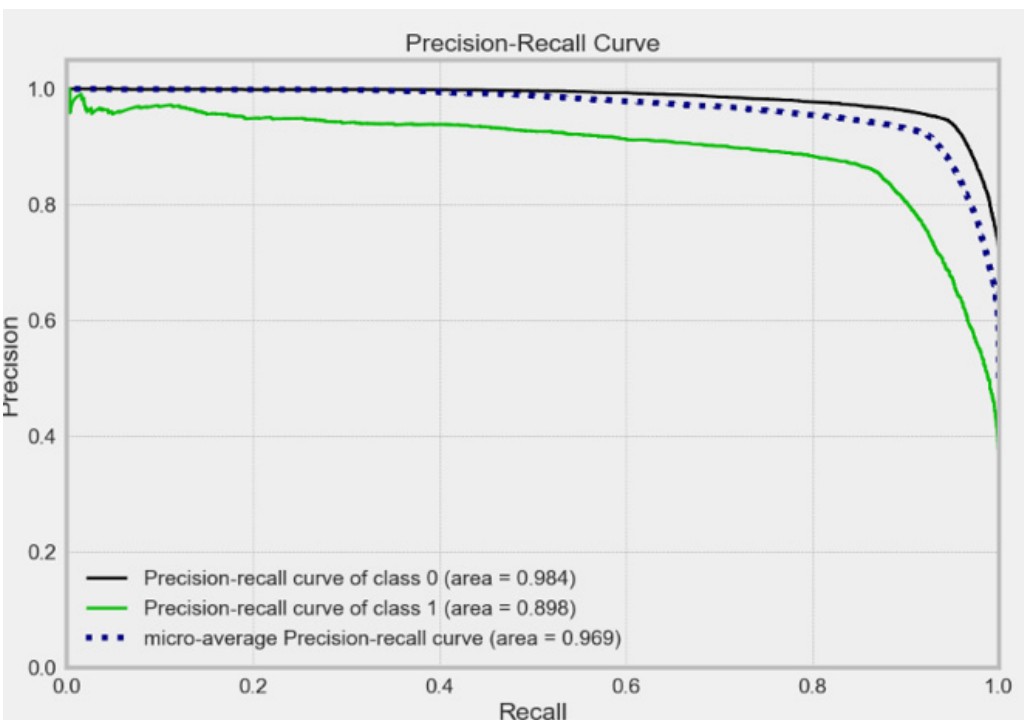

**Figure 20.** Precision–recall Curve for random forest model.

For training the LightGBM model, the following optimal parameters were used: lambda l1 = 1.5, lambda l2 = 1, learning rate = 0.01; min data in leaf = 50; num boost round = 2000; and reg alpha = 0.1. The LightGBM prediction model had an AUC score of 0.9624, an accuracy of roughly 92.23%, a precision of 94.40%, and a recall of 100%. These results are displayed in Table 3 and Figures 10–12.

Max depth = 3, Learning rate = 0, and n estimators = 140 were the best settings we used to train the XGBoost model. The XGBoost prediction model had an AUC score of 0.9623, an accuracy of roughly 92.10%, a precision of 94.58%, and a recall of 100%. These results are displayed in Table 3 and Figures 15–17.

We trained the random forest model with the following ideal parameters: entropy was the criteria, max depth was 8, and n estimators were 120. Table 3 and Figures 16–18 demonstrate the random forest prediction model's accuracy, precision, recall, and AUC score, which were all around 92.18 percent, 93.9%, and 100%, respectively. We chose the following ideal values for the Catboost model's training: depth = 6, learning rate = 0, and number of iterations = 100. The Catboost predictive model had an AUC score of 0.9626, a precision of 94.64%, a recall of 100%, and an accuracy of about 92.23%.

The accuracy, precision, recall, and AUC score of the multilayer perceptron model were all around 91.75 percent, 93.62%, 100%, and 0.9507, respectively. Overall, the models performed about equally well, with CatBoost having the best performance and the neural network model having the poorest, demonstrating that, in some situations, neural networks are not more effective than traditional machine learning algorithms.

The normalized confusion matrix of the CATBoost model is depicted in Figure 21, where 94% of the genuine negative classes were predicted to be negative, and 87% of the real positive classes were predicted to be positive.

The ROC curves for the CATBoost model, which are a graphic representation of the trade-off between the model's sensitivity (i.e., the true positive rate) and its specificity (i.e., the true negative rate), are shown for each class in Figure 22. Moreover, Figure 23 illustrates the precision–recall curves for the CATBoost model for each class, which is a graphical depiction of the model's trade-off between precision and recall.

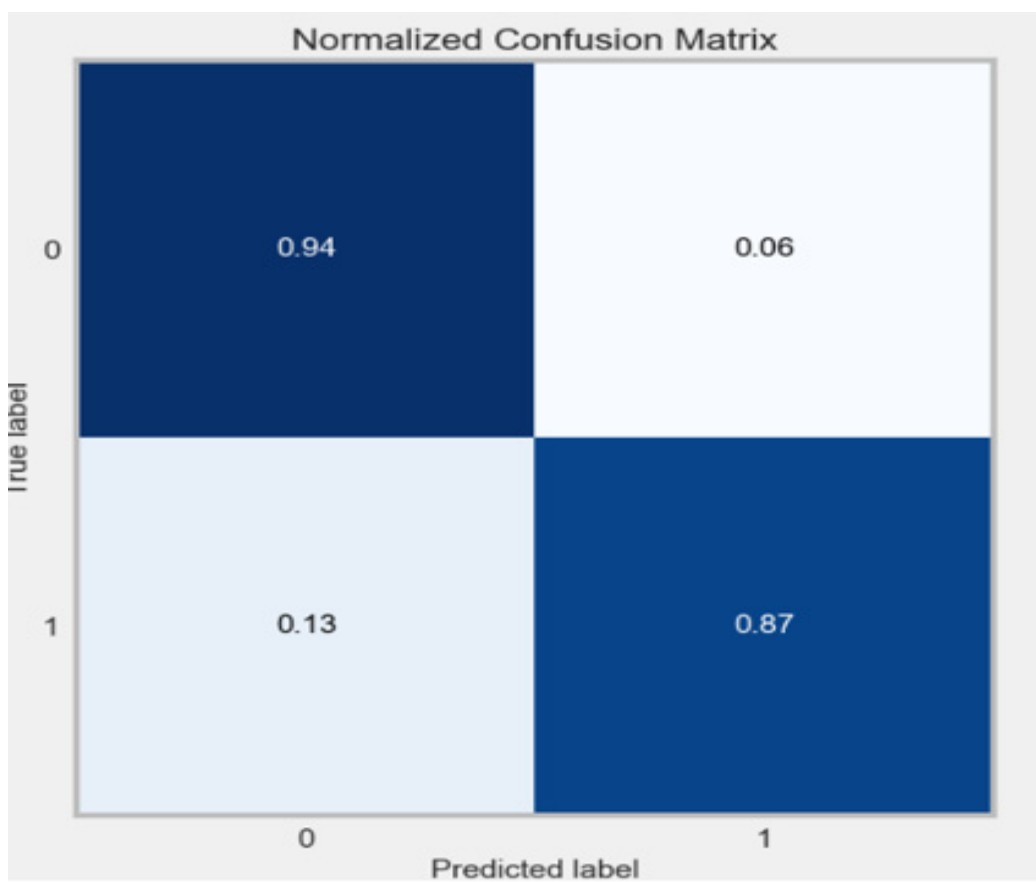

**Figure 21.** Normalized Confusion Matrix for Catboost Model.

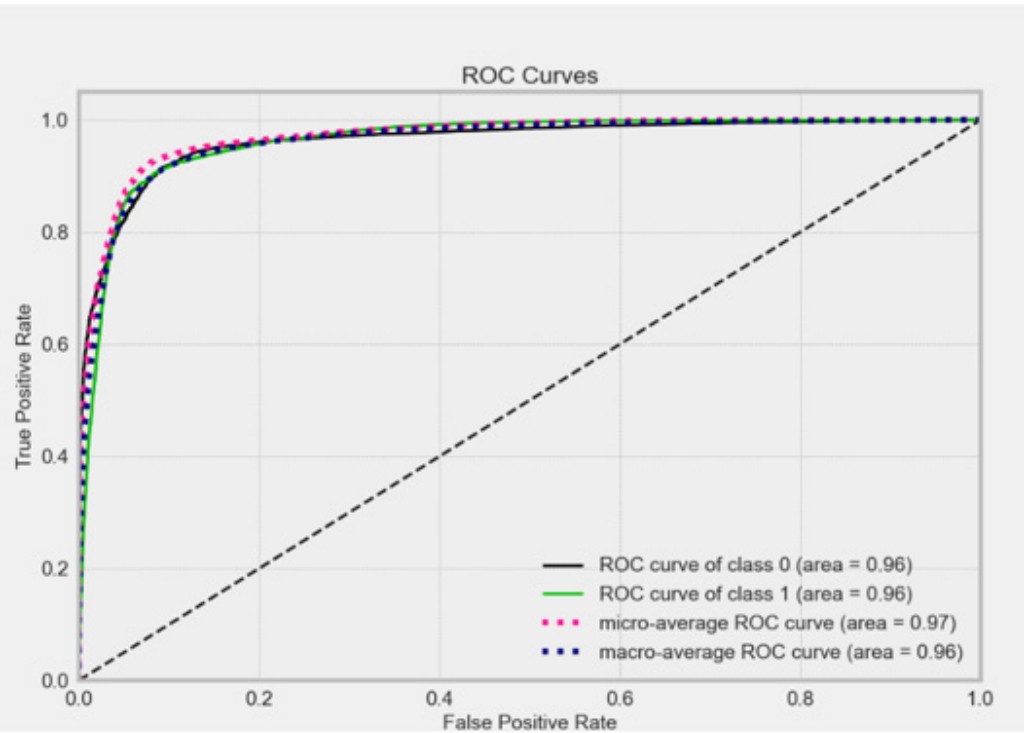

**Figure 22.** ROC Curves for Catboost models.

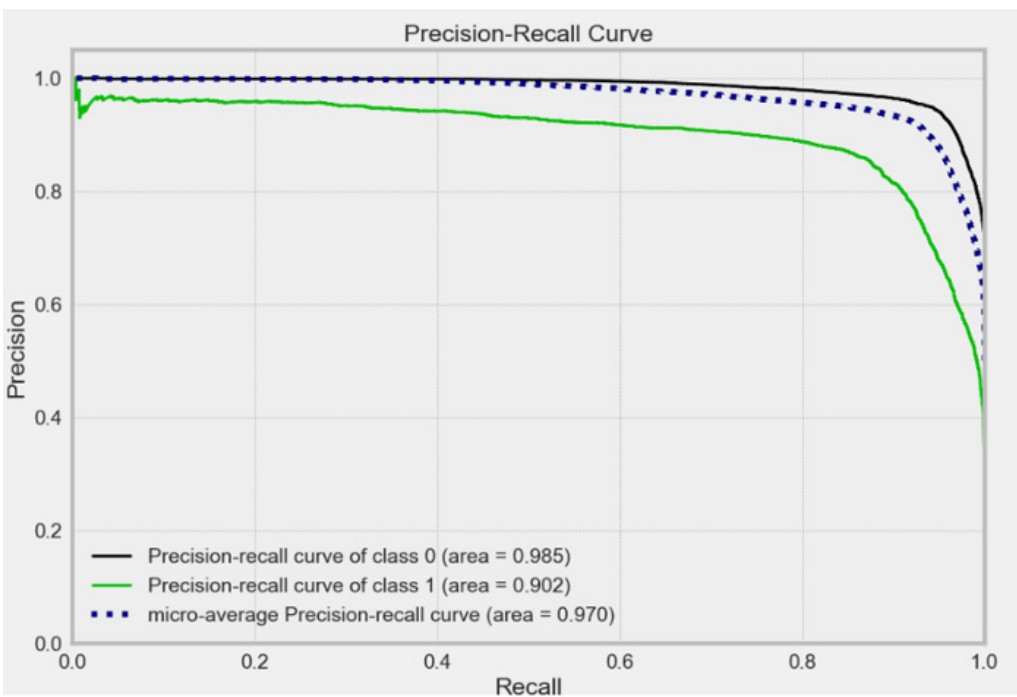

**Figure 23.** Precision–recall Curve for Catboost model.

*5.2. Comparative Analysis*

Table 4 illustrates the accuracy results between our models and the machine learning algorithms used in [19]. Our proposed models perform better compared to the machine learning algorithms used in [19], where the best model is J48 with an accuracy of 88.52% compared with our models' accuracy of around 92.1%.

**Table 4.** Accuracy Comparison.

| Model | Accuracy (%) |
| --- | --- |
| Machine learning models used in [19] | |
| Decision tree | 85.91 |
| J48 | 88.52 |
| Classification and regression tree | 82.22 |
| Gradient-boosting tree | 86.43 |
| Naïve Bayes | 82.93 |
| Proposed model | |
| LightGBM | 92.22 |
| XGBoost | 92.10 |
| Random forest | 92.18 |
| CATBoost | 92.23 |
| Neural network | 91.75 |

Table 5 shows the comparison between the models used in [19], the AISAR model [43] and our models. The AISAR model achieved a precision of 91% compared to our model which reached a precision of 94.64%, which shows that even if the AISAR model has better accuracy, our models return more relevant results, i.e., if our models predict that a student has high engagement, this is correct 94.64% of the time. The recall is also much lower than in our models, with the AISAR model having only around 50% recall (from

the recall–precision curve) compared to ours, which is around 92%, implying that a large fraction of truly positive elements (high engaged students) is captured.

**Table 5.** Comparison between References [19,43] and Our Models.

| Ref | Technique | Accuracy (%) | Precision (%) | Recall (%) |
|---|---|---|---|---|
| [19] | Decision tree | 85.91 | - | 93.40 |
| | J48 | 88.52 | - | 94.7 |
| | Classification and regression tree | 82.22 | - | 89.4 |
| | Gradient-boosting tree | 86.43 | - | 91.0 |
| | Naïve Bayes | 82.93 | - | 90.0 |
| [43] | AISAR model | 97.21 | 91.0 | ~50.0 |
| Proposed Model | LightGBM | 92.22 | 94.4 | 93.21 |
| | XGBoost | 92.10 | 94.58 | 93.2 |
| | Random forest | 92.18 | 93.94 | 92.78 |
| | CATBoost | 92.23 | 94.64 | 93.3 |
| | Neural network | 91.75 | 93.64 | 91.9 |

## 6. Discussion

The current study is concerned with measuring student involvement using technology. The results show that one of the important factors that greatly contributes to forecasting student engagement is the data from the students' VLE activities, which displays how they connect with the environment. It was discovered that the highly involved student took part in these activities more than the less-engaged student. It has been found that the strategies examined work well for spotting academic failure in pupils at an early stage. The study has shown that a machine learning algorithm produces better categorization outcomes. The CATBoost, random forest, XGBoost, and LightGBM models demonstrated greater understanding in predictive analysis when we experimented with several machine learning techniques. From this outcome, it may be inferred that VLE activities and scores have a linear relationship. The classification algorithm will produce better results with diverse features. The aforementioned findings demonstrate that VLE activities by themselves are a potent predictor of student engagement. The XGBoost prediction model had an accuracy of roughly 92.10%, a precision of 94.58%, and a recall of 100%.

The accuracy, precision, recall, and AUC score of the LightGBM prediction model were 92.23%, 94.40%, 100%, and 0.9624, respectively. Precision, accuracy, recall, and AUC score of the Catboost and random forest models were all around 92.18%, 100%, 93.6%, and 0.9606, respectively. The multilayer perpetron model had an accuracy, precision, and AUC score of 93.62%, 100%, and 0.9507, respectively. Moreover, the CatBoost model performed the best overall, while the neural network model performed the poorest, demonstrating that there are some types of issues where classical machine learning algorithms do better than neural networks. Model assessment assesses the model's performance on a set of data in order to gauge how well a machine learning model can forecast future data. It can also enable us to assess how well it predicts students' participation in online learning in comparison to earlier studies. By dividing the total number of positive examples in the dataset by the percentage of accurate positive predictions, the recall is determined. With a high recall score, the model may successfully identify the bulk of the dataset's positive examples, or all the students who demonstrate high engagement. The model's capacity to generate a negligibly small number of erroneous positive predictions is measured by accuracy, recall, precision, and f1. A high precision score, for instance, indicates that the model can produce few or no false positives. The LightGBM prediction model had an AUC score of 0.9624, an accuracy of around 92.23%, a precision of about 94.40%, a recall of 100%, with the area

under the receiver operating characteristic (ROC) curve being used to determine the AUC (area under the curve) score. To train the random forest model and the CatBoost model, we utilized the following ideal training parameters: the accuracy, precision, recall, and AUC score of the random forest predictive model were roughly 92.18%, 93.9%, 100%, and 0.9606, respectively. In addition, CatBoost's accuracy was about 91.75%, precision was about 93.62%, recall was 100%, and AUC was about 0.9507. The normalized confusion matrix of the CATBoost model, which predicted that 94% of the truly negative classes would be negative, and the ROC curves for each class graphically show the trade-off between the model's sensitivity (true positive rate) and its specificity (true negative rate). The precision of the AISAR model was 91%, while our models' precision was 94.64%, showing that our models produce more helpful findings. For instance, our models will be 94.64% accurate when predicting that a student will be highly engaged.

This data-driven study's primary subjects are VLE activities and student evaluation findings. In a follow-up study, we plan to consider additional factors, such as learning preferences and the characteristics of learning materials. In order to determine whether there is a correlation between student participation and grades, we are now examining the relationship between dropout rates and student engagement in VLE. Getting to know students better might lead to better marks and fewer dropouts in any MOOC taking place in online-learning environments.

## 7. Conclusions

Students' frequent lack of enthusiasm for the multiple tasks they must complete in their courses is the biggest problem with e-learning systems. In this work, we employed machine learning (ML) techniques to investigate the effects of poor engagement on students' performance in a social science course at the Open University. The data was gathered using a VLE and processed utilizing a number of data preprocessing techniques, such as the removal of missing values, normalization, encoding, and outlier detection. We employed a range of ML methods to categorize our data, and we assessed each method using cross-validation procedures and a variety of meaningful metrics. GridSearch was used to optimize the ML models in order to find the best-performing hyperparameters. The examination of the models revealed strong prediction performance, with high accuracy, precision, recall, and AUC overall, and with the CATBoost model having a slight edge over the rest. The constructed models also outperformed the models developed in the earlier research in every measurable way. Student involvement is a complicated topic that is influenced by a variety of factors, such as course content, teaching style, and instructor background. The context of student participation requires a deeper examination of these components.

These models are simple to incorporate into VLE systems to support instructors in evaluating student engagement during VLE courses. The precision, accuracy, and recall of the LightGBM prediction model were 92.23%, 94.40%, 100%, and 0.9624, respectively. In order to calculate the receiver operating characteristic (ROC) curve's area under the curve, a formula is used. With an AUC score of 0. 9623, the XGBoost prediction model had an accuracy of roughly 92.10%, a precision of 94.58%, a recall of 100%, and a recall rate of 100%. With an AUC score of 0. 9626, the CatBoost predictive model had an accuracy of roughly 92.23%, a precision of 94.64%, a recall of 100%, and a recall rate of 100%. To train the CatBoost model, the following ideal values were used: depth = 6, learning rate = 0, and n iterations = 100. Last but not least, the CatBoost model outperformed the neural network model, which underperformed overall, showing that there are certain problems in which traditional machine learning methods outperform neural networks. Additionally, the AISAR model achieved a precision of 91% as opposed to our models' precision of 94.64%. The fact that our method was 93.64% accurate in predicting that a student would be very engaged shows that it generates more relevant results.

In future, we intend to examine other aspects of students' engagement analytics in the VLE, such as their learning preferences, the features of their learning materials, and

their reactions to tests and assignments. The association between student engagement and dropout rate in online learning environments can then be observed using the dataset that is produced. Better understanding of students and enhancement of their educational experience can lead to better grades and fewer dropouts in any massive open online courses taking place on virtual platforms.

**Author Contributions:** Conceptualization, N.A. and M.Z.; methodology, M.Z.; software, N.A.; validation, N.A. and M.Z.; formal analysis, N.A.; investigation, M.Z.; resources, M.Z.; data curation, N.A.; writing—original draft preparation, M.Z.; writing—review and editing, N.A.; visualization, N.A.; supervision, N.A.; project administration, N.A.; funding acquisition, N.A. All authors have read and agreed to the published version of the manuscript.

**Funding:** This research is funded by: Researchers Supporting Project number (RSPD2023R608), King Saud University, Riyadh, Saudi Arabia.

**Data Availability Statement:** The dataset is publicly available and can be accessed from this paper: J. Kuzilek, M. Hlosta, and Z. Zdrahal, "Open University learning analytics dataset," Sci. data, vol. 4, no. 1, pp. 1–8, 2017.

**Conflicts of Interest:** The authors declare no conflict of interest.

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
