# Peer review of "Student-Engagement Detection in Classroom Using Machine Learning Algorithm"

_electronics, doi:10.3390/electronics12030731_

Round 1

Reviewer 1 Report

There is a comment early in the paper where it associates expulsion with attention. While there might be some statistically correlation, there is no research indicating that expulsion is directly correlated to attention. Retention is definitely correlated, so that should be adjusted.

There are some formatting issues that should be fixed.

Author Response

Student Engagement Detection in Classroom using Machine Learning Algorithm

Response to the Reviewers

Dear sir/madam

Thank you for giving us the opportunity to submit a revised draft of the manuscript “Student Engagement Detection” for publication. We appreciate the time and effort that you and the reviewers dedicated to providing feedback on our manuscript and are grateful for the insightful comments on and valuable improvements to our paper. We have incorporated most of the suggestions made by the reviewers. Those changes are highlighted/ written in red within the manuscript. Please see below, in blue, for a point-by-point response to the reviewers’ comments and concerns.

Reviewer # 1

Comment 1: There is a comment early in the paper where it associates expulsion with attention. While there might be some statistically correlation, there is no research indicating that expulsion is directly correlated to attention. Retention is definitely correlated, so that should be adjusted.

Author Response: The authors would like to thank the reviewer for this notice. Yes, retention is correlated with attention. We have added a statement in the introduction section making it more relevant to the study.

Comment 2:  There are some formatting issues that should be fixed.

Author Response: Thank you so much for the comment. We have undertaken all the correction related to formatting.

Reviewer 2 Report

This is an interesting paper which examines the prediction of student participation in online courses using machine learning algorithms which analyse VLE data. I think the underlying study is very worthwhile, and the paper could be quite valuable. But in the present version there are some aspects which are unclear and some parts of the paper that I feel need to be expanded or reorganised prior to publication.

In my view, the authors should address the following points:

·         The Introduction needs to more clearly define the meaning of the terms “student engagement” and “participation”, explain the difference between them, and then use the two terms consistently throughout the rest of the paper.

·         The introduction should also define any technical terms that are used, such as behavioural traits, rather than assuming that the reader will understand them.

·         The Introduction should make a clear statement about what the study will add to the literature on predicting student engagement/participation, and why other researchers who work on this topic should find this paper useful. In particular, what is the gap that this paper addresses?

·         In my view the description and diagrams of the model used (Figures 1 and 2) should be separated into a dedicated section, perhaps called Conceptual Structure or something similar. It might also make sense to position this material after the literature review, though this decision can be considered by the authors.

·         The findings are interesting, and the research questions are interesting, but they simply do not match. In my view, the research questions should be changed to match the actual findings presented. Perhaps the RQs should simply be about the accuracy, precision and recall of a predictive model for student participation?

·         The literature review section, and especially the content on pp. 5-6, would be better if more directly addressed to the prediction of student engagement, rather than wider issues of student engagement more generally.

·         Table 1 is useful, but I think the literature review section needs some extra paragraphs afterwards to explain the conclusions being drawn from this analysis. What are the key points from these papers and what work still remains to be done? What will this study add?

·         I think that the ‘dataset’ section would be better if repositioned as a subsection of the Methodology. I also think that the text should draw out the links with figures 1 and 2 more directly.

·         I think that the methodology needs some statement about data access (how this was achieved) and research ethics approval.

·         I think that the methodology needs more clarity about the sequence of steps. Around page 15 I was feeling slightly confused about what had been done and in what order across the project as a whole. Perhaps we need a set of subsections with an overall flowchart?

·         The Results section needs some introductory signposting. What will be presented and in what order?

·         The Results section needs to state some explicit justification for what is presented, in terms of explaining how the results respond to the research questions.

·         My sense is that the results section needs subsections. Inside each subsection you need to group the relevant figures and explain them in text. Some figures don’t have enough explanation, so please be consistent for each. In other cases, the explanation is referring to figures that have not yet been presented, and so the layout needs some re-consideration.

·         The Discussion needs to compare the findings with the studies reviewed earlier—probably those summarised in table 1. What have the authors found that is particularly different, novel or valuable by comparison with those studies?

·         The Conclusions section should reflect on the limitations of this work, and make suggestions for future work that can build on this paper.

Author Response

Student Engagement Detection in Classroom using Machine Learning Algorithm

Response to the Reviewers

Dear sir/madam

Thank you for giving us the opportunity to submit a revised draft of the manuscript “Student Engagement Detection” for publication. We appreciate the time and effort that you and the reviewers dedicated to providing feedback on our manuscript and are grateful for the insightful comments on and valuable improvements to our paper. We have incorporated most of the suggestions made by the reviewers. Those changes are highlighted/ written in red within the manuscript. Please see below, in blue, for a point-by-point response to the reviewers’ comments and concerns.

Reviewer # 2

Comment 1: This is an interesting paper which examines the prediction of student participation in online courses using machine learning algorithms which analyse VLE data. I think the underlying study is very worthwhile, and the paper could be quite valuable. But in the present version there are some aspects which are unclear and some parts of the paper that I feel need to be expanded or reorganised prior to publication.

Author Response: The authors would like to thank the reviewer for this kind, encouraging, and motivating words

In my view, the authors should address the following points:

Comment 2: The Introduction needs to more clearly define the meaning of the terms “student engagement” and “participation”, explain the difference between them, and then use the two terms consistently throughout the rest of the paper.

Author Response: Thank you for pointing this out. The reviewer is correct, and we have added terms definition like student engagement, student participation, cognitive engagement, behavioural traits etc. Also, words like student engagement are being used several times across the paper, e.g in introduction only, we have used it 26 times and same in literature review, we have used it 16 times.

‘Here, the term, ‘student engagement’ describes a measure of a student's level of interaction with others, plus the quantity of involvement in and quality of effort directed toward activities that lead to persistence and completion. Page 1

The term "cognitive engagement" describes the psychological commitment committed to educational activities [7][8], where the learner is motivated to learn. Page 2.

The term ‘student participation’ describes it as an assessment of a student’s performance in a course outside of their assessments. Here the items that might be evaluated in student participation are engagement in a class discussion, engagement in online discussion and student behaviour in group settings. Page 2’

Comment 3: The introduction should also define any technical terms that are used, such as behavioural traits, rather than assuming that the reader will understand them.

Author Response: Thank you for pointing this out. The reviewer is correct, and we have added terms like behavioural traits. It is now updated in the paper.

‘The term ‘behavioural traits’ in student engagement refers to the observable act of students who are being involved in learning. It also refers to students’ participation in academic activities and efforts to perform on different tasks. Page 3.’

Comment 4: The Introduction should make a clear statement about what the study will add to the literature on predicting student engagement/participation, and why other researchers who work on this topic should find this paper useful. In particular, what is the gap that this paper addresses?

Author Response: We would really like to thank the reviewer for this question. Here in this work we have presented a new methodology to handle the issue of student engagement detection in the class. This study is unique as it can be seen below. This content is highlighted in the introduction section of the paper.

“This study uses extracted raw data to compute and describe student engagement. The most common data preprocessing used in engagement detection models in this field includes converting and formatting the input data, normalization and encoding is another technique used in this paper, followed by predicting and developing models like XGBoost [20], LightGBM [21], Random Forest [22], CatBoost[23] and finally the Stacking technique. Here, we create a deep-learning model with two crucial steps—basic data input detection and engagement recognition—to overcome the is-sues. To create a rich input representation model and achieve cutting-edge perfor-mance, a convolutional neural network (CNN) is trained on the dataset in the first step. This model is used to initialize our engagement recognition model, which was created using a different CNN and was trained using our recently amassed dataset in the engagement recognition domain.”

Comment 5: In my view the description and diagrams of the model used (Figures 1 and 2) should be separated into a dedicated section, perhaps called Conceptual Structure or something similar. It might also make sense to position this material after the literature review, though this decision can be considered by the authors.

Author Response: Thank you for pointing this out. The reviewer is correct. We have moved the respective figure 2 to the literature review section as it fits well in this section and also, I have written the figure description. Whereas, figure 1 remains in the introduction section. And same figure description is now given.

‘Figure 1: Conceptual Structure for Student Engagement Using ML/DL Methods in a Classroom. Page 3. Remains in the Introduction section.

             Description: The conceptual framework for student participation is shown in Figure 1. Data gathering and analysis come first. Data pre-processing, feature selection, classifiers, and having all the datasets are followed by data transformation and classification. The pupils are then divided into high- and low-engagement students by the evaluating and testing procedure, which employs a variety of techniques and a prediction model.’

‘Figure 2: Techniques applied in different papers related to Student Engagement. Page 6 of Literature Review.

Figure 2 shows the methods used in various studies on how to keep students engaged in the classroom. In these studies, student extraction takes the form of task data, actions, rating comments, and statistics. Also included are XGboosy, LightGBM, Random Forest, neural networks, CATBoost, and stacking methods as machine-learning or deep-learning models. Finally, we receive the anticipated outcome. The model evaluation also includes the MOOC, VLE, CNN, EM, and AISAR Models. The lecturer and students are then given the results.’

Comment 6: The findings are interesting, and the research questions are interesting, but they simply do not match. In my view, the research questions should be changed to match the actual findings presented. Perhaps the RQs should simply be about the accuracy, precision and recall of a predictive model for student participation?

Author Response: Thank you for pointing this out. The reviewer is correct. See the latest addition of questions added to get more clearer view, what we are going to discuss in the result section. Updated as per the requirement.

‘In order to forecast student engagement and participation in a VLE, this study compares various machine learning algorithms. The teachers can intervene in the learning process even at an early stage of the courses by using the accurately projected results. It is challenging for the models to provide accurate predictions due to the abundance of model parameters and the reliability of their properties. Therefore, the current study places more emphasis on the variables that help with the early prediction of the VLE engagement criterion. The project also seeks to identify the best classifier that can handle the varied, heterogeneous data from the VLE log. The following research queries are addressed in this work:

Which classifier performs best in predicting students' VLE engagement?

What are the variables that affect a VLE's ability to forecast student engagement optimally?

Page 4 of the Introduction.

Comment 7: The literature review section, and especially the content on pp. 5-6, would be better if more directly addressed to the prediction of student engagement, rather than wider issues of student engagement more generally.

Author Response Thank you for pointing this out. The reviewer is correct. Updated as per the comments made related to the addressing to the prediction of student engagement. Whereas, the paragraphs which were not needed, are removed from the literature review section.

See page 5, written in red. ‘’ Similar to this, Goldberg et al. [32] employed a variety of ML methods to analyse the Student Performance Dataset and the Student Academic Performance Dataset, where the former dataset was used for prediction and the latter for categorization. Students' online behaviours were taken into consideration by Hastings et al. [33] to forecast their success while utilising an e-learning system. Using information gleaned from students' log-in histories and learning management system on the Sakai platform, the author classified students according to their learning styles [33]. Preprocessing, feature selection, and parameter optimization were done before classification. The performance of students in a particular course can be predicted with the use of this kind of categorisation. Another study [34] shown that ML methods may accurately predict a student's final grades by using their prior grades. A dashboard was created to predict students' engagement and performance in real-time, which could help stop students from deciding to drop out too early. In a different study [35], ML methods were employed to predict students' engagement based on their behavioural characteristics and to examine how evaluation marks were affected. With the use of a dashboard that shows students' behaviours in the learning environment, instructors can quickly detect pupils who aren't paying attention in class. An adaptive gamified learning system [32–34] was created to boost student engagement in the learning environment and, as a result, performance. It integrates educational data mining with gamification and adaption approaches. In the context of online learning, the effectiveness of gamification in comparison to adaptive gamification was examined. Three classifiers were used by Wolff et al. [34] to create a framework for predicting student engagement and performance. By eliminating unnecessary and duplicate elements, the authors preprocessed the data obtained from the Kalboard 360 online learning management system. They then carried out feature selection and analysis to determine which features were the most discriminating. Finally, classification techniques were employed to forecast the performance of the students.’’

See page 6, written in red. ‘Larsen [43] applied statistical methods to predict student engagement in a web-based learning environment and concluded that variables such as course design, teacher participation, class size, student gender, and student age need to be controlled for when assessing student engagement. Manwaring et al. [43] conducted a study to understand student engagement in higher education blended-learning classrooms. This study used a cross-lagged modeling technique and found that course design and student perception variables greatly affected student engagement in the course. Aguiar et al. [44] conducted a study to investigate the relationship between a student’s final score and the student’s engagement in material using a statistical technique and found that students who had high levels of engagement for quizzes and materials earned higher grades on the final exam. Thomas et al. [45] developed an early-warning system using engagement-related input features and found that these variables are highly predictive of student-retention problems. Atherton and Shah [46] measured student engagement using an ML algorithm based on students’ facial expressions, head poses, and eye gazes. Their results showed that ML algorithms performed well at predicting student engagement in class. Bosch et al. [47] found a correlation between the use of course materials and student scores; students who accessed course content more often achieved better results on their exams and assessments. Piech [48] studied the automatic detection of student cognitive engagement using a face-based approach.’

Comment 8: Table 1 is useful, but I think the literature review section needs some extra paragraphs afterwards to explain the conclusions being drawn from this analysis. What are the key points from these papers and what work still remains to be done? What will this study add?

Author Response : Thank you for pointing this out. The reviewer is correct. Kindly see the updated section written after the table 1. Different models which were used by the researchers, are all mentioned in the table. In the written paragraph, we have mentioned about our model.

‘Numerous works in this area of student participation have been produced, according to various authors. The most common learning environment is a physical classroom or an online classroom. To extract variables like student performance evaluation, student engagement, cognitive engagement, and student performance from the datasets, various techniques were used, and each of these approaches produced results with a varying level of accuracy. In these publications, writers compared their models to earlier models created by other authors, and in most cases, their models' accuracy outperformed their competitors'. Similarly, in this work, our models demonstrated strong prediction performance with high accuracy, precision, recall, and AUC scores overall, with a little edge for the CATBoost model. Our created model beat earlier studies in every way. In this study, we contrast several models, including XGBoost, RF, MLP, and lastly the AISAR model, which was created with greater overall accuracy.’ See page 8.

Comment 9: I think that the ‘dataset’ section would be better if repositioned as a subsection of the Methodology. I also think that the text should draw out the links with figures 1 and 2 more directly.

Author Response: Thank you for this suggestion. As suggested by the reviewer, we have revised/ repositioned dataset as subsection of methodology.

Comment 10: I think that the methodology needs some statement about data access (how this was achieved) and research ethics approval.

Author Response: Thank you for this suggestion. As suggested by the reviewer, we have revised the opening paragraph with figure 3 added with writeup. Please see page 9 and10.

OULAD Framework figure 3- ‘The dataset contains the following seven parameters: StudentInfo, StudentAssessment, Assessment, StudentVLE, StudentRegistration, VLE, and Courses. CSV files with these parameters are accessible. Figure 3 displays the parameters' for OULAD framework. Data on the students' demographics, registration, assessments, and VLE interactions are included in the dataset. In the assessment table, the learner performance is broken down into four categories: distinction, pass, fail, and withdrawn. Each student's results and accomplishments over the course of the course are reflected in their performance as a learner. Utilizing interaction data from the VLE, the current study concentrates on the level of student engagement.

It includes evaluations from module presentations that are followed by final exams as well as information on seven chosen courses, which are referred to as modules in the dataset. The dataset also contains information about the student's location, age, disability, education level, gender, and other factors. Additionally, results from student evaluations and their interactions with the online learning environment are given (VLE). The first step is collecting and analysing data. Data transformation and classification come after data pre-processing, feature selection, classifiers, and possessing all the datasets. Then, using a number of tools and a prediction model, the evaluating and testing procedure separates the students into high- and low-engagement kids.’

Comment 11: I think that the methodology needs more clarity about the sequence of steps. Around page 15 I was feeling slightly confused about what had been done and in what order across the project as a whole. Perhaps we need a set of subsections with an overall flowchart?

Author Response: Thank you for this suggestion. As suggested by the reviewer, we have revised few lines on page 11 about dataset.

‘Raw databases are often not in a form that is suitable for analysis, so it is common to perform some initial preparation on the data before proceeding with further analysis. The dataset used for the predictive model was cleaned of unnecessary attributes, which will not impact the prediction of the output, such as the identification of the student, the site, the assessments, and the presentations, and other irrelevant attributes such as the length of the course, the type and duration of the assessments.

The final dataset contains 13 features and 32593 observations, with 8 features being categorical and 5 being numerical. This table 2 shows the features selected for the training of the model used:

Also, kindly see the figure 6 and writeup. I have created one overall figure and writeup for the sequence of steps. See page 13 and 14.

‘The flowchart for the entire list of actions needed for this method is shown in Figure 6. To begin with, we create the best prediction model for categorising student activity in a VLE. Following data pre-processing, OULAD is measured using the learning environment. It is separated into normalisation approaches and missing values. The classifier and feature selection are the results of data pre-processing. A predictive model is then produced throughout the evaluation and testing phases.’

Comment 12: The Results section need some introductory signposting. What will be presented and in what order?

Author Response: Thank you for this suggestion. As suggested by the reviewer, we have revised few introductory lines for the result section.

‘The goal of the project is to create a model that can forecast a student's level of engagement in a VLE, from high to low. A portion of the OULAD dataset is used for the investigation, and the characteristics are chosen using correlation measurement. The dataset was created by pulling out and combining various columns from seven tables that were made accessible as OULAD. The resulting dataset has a row for each student. According to the assessment score considered, as described in the data pretreatment section, the rows are then marked as highly or lowly engaged pupils. The earlier stated algorithms have already shown a correlation between evaluation scores during the course and final scores with engagement.’ See page 18

Comment 13: The Results section needs to state some explicit justification for what is presented, in terms of explaining how the results respond to the research questions.

Author Response:  Yes, you are right, and I have updated in accordance with the needs. We have finally addressed the research questions which we mentioned in the introduction and its explanation is done on page 18 and 19.

‘Which classifier performs the best in order to predict student participation in a VLE? It is the very first question that was posed in the introduction section, and it is the focus of this result section. We can see that a variety of data sources were used to create the predictive models. The assessment result, the end result, and the students' selection of VLE activities are the features in this case. The degree of student engagement is the anticipated variable iIn Table 2, evaluation metrics for various models were also compiled.

Likewise, second question which is given in the introduction part, focused on the Student engagement with the homepage, outgoing content, subpage, URL, and forum is a subset of activities that exhibits more encouraging outcomes. The result reported here is consistent with past research using datasets like OULAD and others. When compared to a student who was less engaged, a highly engaged student was found to interact with these activities more. Additionally, the writers were able to find a correlation between the results of these activities' clicks and their scores. In order to anticipate at-risk pupils, it is possible to use the interaction with these activities. Recall is the main metric used to determine which students are least involved, and accuracy is used to assess how well the model predicts which students will be most engaged.’

Also note that all the figures and tables are fully cited and marked in red color on the main file.

See page 19: ‘Above figure 12 shows the Normalized confusion matrix for Light GBM Model. Whereas, figure 13 depicts the ROC curve for LightGBM models.’

See page 20: ‘Above figure 14 depicts Precision-recall curve for LightGBM model. Whereas, figure 15 shows the Normalized confusion matrix for XGBoost Model.’

And So on.

See page 21, 22, 23, 25, 26. All the figures are cited with description as well.

Comment 14: My sense is that the results section needs subsections. Inside each subsection you need to group the relevant figures and explain them in text. Some figures don’t have enough explanation, so please be consistent for each. In other cases, the explanation is referring to figures that have not yet been presented, and so the layout needs some re-consideration.

Author Response: Also note that all the figures and tables are fully cited and marked in red color on the main file.

See page 19: ‘Above figure 12 shows the Normalized confusion matrix for Light GBM Model. Whereas, figure 13 depicts the ROC curve for LightGBM models……………..’

See page 20: ‘Above figure 14 depicts Precision-recall curve for LightGBM model. Whereas, figure 15 shows the Normalized confusion matrix for XGBoost Model……………….’

And So on.

See page 21, 22, 23, 25, 26. All the figures are cited with description as well.

Even the table 4 and 5 are cited and description is written.

On page 25 and 26, we have created tables to give you more insight, what we have discussed in the result section. Different models with their accuracy rates are all given in the table. It not only compared the proposed model but also the models/ classifiers we have mentioned in table 1. Pleae see table 4 and 5.

Comment 15: The Discussion needs to compare the findings with the studies reviewed earlier—probably those summarised in table 1. What have the authors found that is particularly different, novel or valuable by comparison with those studies?

Author Response: We appreciate the reviewer’s feedback, but in discussion section we have compare the findings of different classifiers/ models. See page 27.

‘The current study is concerned with measuring student involvement using technology. The results show that one of the important factors that greatly contributes to forecasting student engagement is the data from the students' VLE activities, which displays how they connect with the environment. It was discovered that the highly involved student took part in these activities more than less engaged student. It has been found that the strategies examined work well at spotting academic failure in pupils at an early stage. The study concluded because it showed that a machine learning algorithm produces better categorization outcomes.’

‘The aforementioned findings demonstrate that VLE activities by themselves are a potent predictor of student engagement. The XGBoost prediction model had an AUC score of roughly 92.10%, a precision of 94.58%, a recall of 100%.’

‘The accuracy, precision, recall, and AUC of the LightGBM prediction model were 92.23%, 94.40%, 100%, and 0.9624 respectively. Precision, accuracy, recall, and AUC score of the Catboost and Random Forest models were all around 92.18%, 100%, 93.6%, and 0.9606 respectively. The Multi-layer Pereptron Model had an accuracy, precision, 93.62%, 100% and. 0.9507 respectively.’

‘The LightGBM prediction model had an AUC score of 0.9624, an accuracy of around 92.23%, a precision of about 94.40%, a recall of 100%, and the area under the receiver operating characteristic (ROC) curve is used to determine the AUC (Area Under the Curve) score. To train the Random Forest model and the Catastrophe model, we utilised the following ideal training parameters: The accuracy, precision, recall, and AUC score of the Random Forest predictive model were roughly 92.18%, 93.9%, 100%, and 0.9606 respectively. In addition, CatBoost's accuracy was about 91.75%, precision was about 93.62%, recall was 100%, and AUC was about 0.9507.’

‘The normalised confusion matrix of the CATBoost model, which predicted that 94% of the truly negative classes will be negative, The ROC curves for each class graphically show the trade-off between the model's sensitivity (true positive rate) and specificity (true negative rate). The precision of the AISAR Model is 91%, while our model's precision is 94.64%, showing that our model produces more helpful findings. For instance, our model will be 94.64% accurate when predicting that a student will be highly engaged.’

Comment 16: The Conclusions section should reflect on the limitations of this work, and make suggestions for future work that can build on this paper.

Author Response: We appreciate the reviewer’s feedback, and as per the comment, we have added the future work and suggestion.

Students' frequent lack of enthusiasm for the multiple tasks they must complete in the courses is the biggest problem with e-learning systems. In this work, we employed machine learning (ML) techniques to investigate the effects of poor engagement on students' performance in a social science course at the Open University. The data was gathered using a VLE and processed utilising a number of data pre-processing techniques, such as the removal of missing values, normalisation, encoding, and outlier detection.

‘In future, we intend to examine other aspects of students' engagement analytics in the VLE, such as their learning preferences, the features of learning materials, and their reactions to tests and assignments. The association between student engagement and dropout rate in online learning environments can then be observed using the dataset that is produced. Better student comprehension and enhancement of their educational experience can lead to better grades and fewer dropouts for any massive open online courses taking place on virtual platforms.’ See page 28

Reviewer 3 Report

Dear authors

Student Engagement Detection in Classroom using Machine 2 Learning Algorithm

In this paper, the authors presented Student engagement is a flexible, complicated concept that includes behavioural, emo- 11 tional, and cognitive involvement. In order for the instructor to understand how the student inter- 12 acts with the various activities in the classroom, it is essential to predict their participation. 

I have some comments that are given as

. Introduction needs improvement

. Which type of data is developed and explored 

. How much data is used for training

. For the optimization, which algorithm is applied? Is it global search method or local search scheme?

. Finally how did you evaluate the model? explain

. Can we use other activation function without sigmoidal function?

. Provide more discussions of the figures

. Update the paper with the latest recent artificial neural network references of Prof. Mohamed Reda Ali (https://scholar.google.com/citations?user=bjNjsmoAAAAJ&hl=ar)

Author Response

Student Engagement Detection in Classroom using Machine Learning Algorithm

Response to the Reviewers

Dear sir/madam

Thank you for giving us the opportunity to submit a revised draft of the manuscript “Student Engagement Detection” for publication. We appreciate the time and effort that you and the reviewers dedicated to providing feedback on our manuscript and are grateful for the insightful comments on and valuable improvements to our paper. We have incorporated most of the suggestions made by the reviewers. Those changes are highlighted/ written in red within the manuscript. Please see below, in blue, for a point-by-point response to the reviewers’ comments and concerns.

Reviewer # 3

Dear authors

Student Engagement Detection in Classroom using Machine 2 Learning Algorithm

In this paper, the authors presented Student engagement is a flexible, complicated concept that includes behavioural, emotional, and cognitive involvement. In order for the instructor to understand how the student inteacts with the various activities in the classroom, it is essential to predict their participation. 

I have some comments that are given as

Comment 1: Introduction needs improvement.

Author Response: Yes, reviewer is correct and we have added definitions, changed/ moved the figures in relevant section. Figures are properly cited and research queries are addressed properly.  Kindly see the Introduction section in the main file. Written in red color.

Comment 2: Which type of data is developed and explored 

Author Response: The dataset includes information on seven chosen courses (referred to as modules in the dataset), assessments given during module presentations that are followed by final exams. The dataset also contains information on the student's location, age group, disability, degree of education, gender, etc.

Additionally included are student assessment scores and interactions with the Virtual Learning Environment (VLE).

Comment 3: How much data is used for training

Author Response: We appreciate the reviewer’s feedback. Yes, please see page 11, ‘it shows the dataset contains contains 32593 observations, where 80% was used for training and 20% is used for testing. i.e 26 074 data rows was used for the training and 6519 for the testing.

Comment 4: For the optimization, which algorithm is applied? Is it global search method or local search scheme?

Author Response: Global search method is applied for the optimization since the global optimum might be impossible to find, and we will have a local optimum. Also, its often hard (impossible) to get the global optimum and you often just get a local optimum that has good performances overall

Comment 5: Finally how did you evaluate the model? Explain

Author Response: We appreciate the reviewer’s feedback and as the reviewer can see that. To evaluate the model, several metrics were used for this purpose other than the log loss metric, such as the accuracy, recall, precision, and f1 score. The recall is the number of true positive predictions divided by the total number of positive examples in the dataset. A high recall score means that the model can successfully recognize most of the dataset's positive examples, i.e., all the students that have a high engagement. The precision is the number of true positive predictions divided by the total number of positive predictions made by the model. A high precision score mean at the model is able to make very few false positive predictions.

Another very important metric in model evaluation that is also used to compare the performances of different models is the AUC. The AUC (Area Under the Curve) score is calculated as the area under the curve of the receiver operating characteristic (ROC) curve, which plots the true positive rate (TPR) against the false positive rate (FPR) at various classification thresholds. A high AUC score means that the model is able to distinguish between positive and negative classes very well.

See from page 19 to 26. We have already explained all in detail, which model gives higher accuracy compared to others.

Comment 6: Can we use other activation function without sigmoidal function?

Author Response:  We can use other activation function other than sigmoidal function in the last predictive layer for the classification, such as the tanh function which is very similar to the sigmoid activation function, and even has the same S-shape with the difference in output range of -1 to 1, instead of 0 to 1 range of the sigmoid function.

Comment 7: Provide more discussions of the figures

Author Response: Thank you for pointing this out. The reviewer is correct. Noted and Updated as per the requirement. You can see that each figure and table is already cited and description is also given with respect to the diagram/ figures/ tables.

Reviewer 4 Report

The paper proposes comparisons of Machine/Deep Learning algorithms to predict student engagement in the classroom.

Overall, the paper is interesting and wants to solve a real problem. However, to improve the quality of the work, I suggest the following improvements.

It is suggested that the state-of-the-art include work that predicts student engagement in the classroom using neuro-fuzzy algorithms that return the explanation of the result in addition to the prediction result like: https://doi.org/10.1007/978-3-030-96308-8_124.  

On page 3 in line 105, delete the space between the words "learning" and "deep."

It is suggested that Figures 3, 4, 9, 10, 11, 12, 17, 18, 19, 20, and 21 should be redone because they are of low quality.

Also, Figures 13, 14, 15 and 16 are mentioned in the text but are not shown. In contrast, figures 17, and 18 are shown but are not mentioned in the text.

In addition, to have higher readability in the text, it is suggested that the figures be merged.

Finally, the authors should state the experimental setup used (Hold-out or Cross-validation).

Author Response

Student Engagement Detection in Classroom using Machine Learning Algorithm

Response to the Reviewers

Dear sir/madam

Thank you for giving us the opportunity to submit a revised draft of the manuscript “Student Engagement Detection” for publication. We appreciate the time and effort that you and the reviewers dedicated to providing feedback on our manuscript and are grateful for the insightful comments on and valuable improvements to our paper. We have incorporated most of the suggestions made by the reviewers. Those changes are highlighted/ written in red within the manuscript. Please see below, in blue, for a point-by-point response to the reviewers’ comments and concerns.

Reviewer # 4

The paper proposes comparisons of Machine/Deep Learning algorithms to predict student engagement in the classroom.

Overall, the paper is interesting and wants to solve a real problem. However, to improve the quality of the work, I suggest the following improvements.

Comment 1: It is suggested that the state-of-the-art include work that predicts student engagement in the classroom using neuro-fuzzy algorithms that return the explanation of the result in addition to the prediction result like: https://doi.org/10.1007/978-3-030-96308-8_124.  

Author Response: We have included this work in the state-of-the-art section with ref [49].

 Casalino, Gabriella, Giovanna Castellano, and Gianluca Zaza. "Neuro-Fuzzy Systems for Learning Analytics." In International Conference on Intelligent Systems Design and Applications, pp. 1341-1350. Springer, Cham, 2022.

Comment 2: On page 3 in line 105, delete the space between the words "learning" and "deep."

Author Response: Thank you for pointing this out. The reviewer is correct. It is updated.

Comment 3: It is suggested that Figures 3, 4, 9, 10, 11, 12, 17, 18, 19, 20, and 21 should be redone because they are of low quality.

Author Response: Reviewer is correct and sometime when you convery docx to pdf, this can happen. But now these figures are magnified and can be easily seen. On the docx file, these figures can be read properly.

Just a example of figure shown below:

See this figure was small in previous version, now it has been magnified so that it can be seen properly on the PDF.

Comment 4: Also, Figures 13, 14, 15 and 16 are mentioned in the text but are not shown. In contrast, figures 17, and 18 are shown but are not mentioned in the text.

Author Response: All the figures are properly cited with good explanation. For you to be comfortable, I have written them in red in the main file.

‘Above figure 12 shows the Normalized confusion matrix for Light GBM Model. Whereas, figure 13 depicts the ROC curve for LightGBM models. The best values for the training parameters for the LightGBM model were: lambda l1 = 1.5, lambda l2 = 1, learning rate = 0.01; min data in leaf = 50; num boost round = 2000; and reg alpha = 0.1. The LightGBM prediction model had an AUC score of 0.9624, a precision of 94.40%, a recall of 100%, and an accuracy of around 92.23%.’

‘Above figure 14 depicts Precision-recall curve for LightGBM model. Whereas, figure 15 shows the Normalized confusion matrix for XGBoost Model. The optimum parameters we used to train the LightGBM and XGBoost model were Max depth = 3, Learning rate = 0, and n estimators = 140.’

Above figure 16 shows ROC Curve for XGBoost model. Whereas, figure 17 depicts the Precision-recall curve for XGBoost model. The XGBoost prediction model had an AUC score of 0.9623, a precision of 94.58%, a recall of 100%, and an accuracy of around 92.10%.

Comment 5: In addition, to have higher readability in the text, it is suggested that the figures be merged.

Author Response: Reviewer is correct but figures and technical writing can be properly understood as you go through the research paper. Figures cannot be merged as merging them will cause trouble to the image quality.

Comment 6: Finally, the authors should state the experimental setup used (Hold-out or Cross-validation).

Author Response: Cross-validation technique is used in this paper. Also, it is clearly mentioned in the paper as well that we are using the cross-validation technique.

‘we assessed each one using cross-validation methods and many helpful indicators.’

Also, it is higlighted in yellow color on the main file. Page 1.

Round 2

Reviewer 3 Report

The paper can be accepted in its present form